# Bidirectional Representations Augmented Autoregressive Biological Sequence Generation

**Xiang Zhang**[1,2,†*]   **Jiaqi Wei**[3,4*]   **Zijie Qiu**[1,3]   **Sheng Xu**[1,3]   **Zhi Jin**[3]

**Zhiqiang Gao**[3]   **Nanqing Dong**[3]   **Siqi Sun**[1,3]

[1] Fudan University   [2] University of British Columbia

[3] Shanghai Artificial Intelligence Laboratory   [4] Zhejiang University

xzhang23@ualberta.ca, siqisun@fudan.edu.cn

## Abstract

Autoregressive (AR) models, common in sequence generation, are limited in many **biological tasks** like *de novo* peptide sequencing and protein modeling by their unidirectional nature, **failing to capture crucial global bidirectional token dependencies**. Non-Autoregressive (NAR) models offer holistic, **bidirectional** representations but face challenges with generative coherence and scalability. To transcend this, we propose a hybrid framework enhancing AR generation by dynamically integrating rich contextual information from non-autoregressive mechanisms. Our approach couples a *shared input encoder* with two decoders: a non-autoregressive one learning latent bidirectional biological features, and an AR decoder synthesizing the biological sequence by leveraging these bi-directional features. A *novel cross-decoder attention* module enables the AR decoder to iteratively query and integrate these bidirectional features, enriching its predictions. This synergy is cultivated via a tailored training strategy with *importance annealing* for balanced objectives and *cross-decoder gradient blocking* for stable, focused learning. Evaluations on a demanding 9-species benchmark of *de novo* peptide sequencing task show our model substantially surpasses AR and NAR baselines. It uniquely harmonizes AR stability with NAR contextual awareness, delivering robust, superior performance on diverse downstream data. This research advances biological sequence modeling techniques and contributes a novel architectural paradigm for augmenting AR models with enhanced bidirectional understanding for complex sequence generation. Our code is available in GitHub.

## 1   Introduction

Biological sequences—including DNA, RNA, and proteins—encode the fundamental information of life [35, 49]. Modeling these sequences requires capturing not only local motifs but also **global bidirectional dependencies**, as distant tokens and elements often interact in ways that critically determine biological function [1, 23, 14]. For example, amino acid residues may form structural or functional interactions across long sequence distances from both directions, and motifs in DNA/RNA frequently depend on upstream and downstream context [51]. Thus, methods limited to unidirectional modeling struggle to fully capture the biological reality where sequence meaning arises from **bidirectional contextual information**.

Autoregressive (AR) models have become a dominant paradigm in biological sequence generation due to their capacity for tractable sequential modeling [14]. However, their inherent unidirectionality imposes a fundamental constraint, limiting their ability to capture the global sequence semantics

---

*Equal contributions. † Work done while interning at Fudan University.

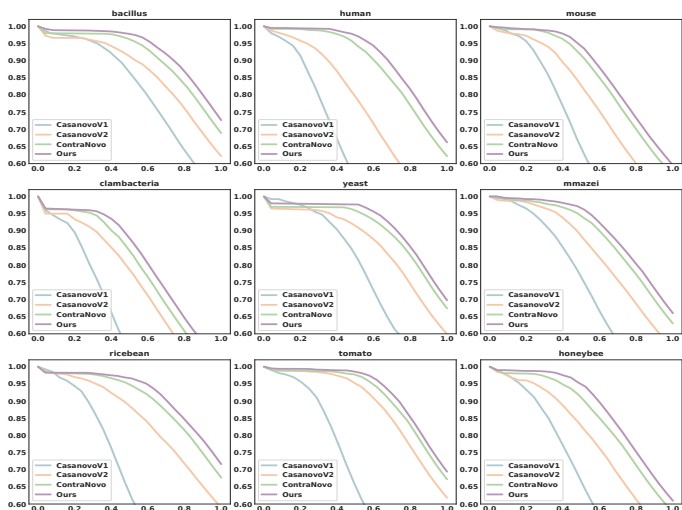

Figure 1: **Peptide Precision-Coverage Curves for Various Species**. Across all subplots, the green lines are consistently positioned above the blue and orange lines, illustrating the superior performance of our model in peptide recall over varying coverage levels. (X-axis: coverage of peptides according to confidence scores, Y-axis: Peptide recall)

often crucial for accurately interpreting biological sequences. In contrast, Non-Autoregressive (NAR) Transformers (NATs) employ holistic bidirectional self-attention mechanisms, enabling them to learn potent bidirectional representations. This characteristic allows NATs to better reflect underlying biological realities and potentially achieve superior performance [46]. Despite these representational advantages, NATs introduce practical and theoretical challenges. Notably, managing variable sequence lengths often necessitating the use of auxiliary length predictors that can be prone to errors [19], navigating complex bi-directional optimization landscapes (e.g., using NAT-specific loss [10, 11]) can lead to training instability and reduced scalability compared to their AR counterparts [26]. These inherent difficulties have, to date, hindered the full realization of NATs' bidirectional processing strengths in the domain of biological sequence modeling.

To transcend the prevailing dichotomy between the generative stability of AR models and the rich contextual understanding afforded by NAR, we introduce CROSSNOVO, a novel hybrid framework. This architecture empowers autoregressive generation by dynamically integrating potent contextual information derived from non-autoregressive mechanisms. Our approach couples a *shared encoder* for input condition encoding, with two specialized decoders for sequence decoding. The first, a non-autoregressive decoder, is tasked with learning latent bidirectional contextual features directly from the input. Concurrently, an autoregressive decoder synthesizes the sequence, its generation process critically informed by these contextual features. The cornerstone of this integration is a novel cross-decoder attention module, which enables the autoregressive decoder to iteratively query and incorporate the rich bidirectional representations captured by its non-autoregressive counterpart. This symbiotic relationship is cultivated through a training strategy incorporating *importance annealing* for balanced multi-objective optimization and *cross-decoder gradient blocking* to ensure focused representational learning and maintain training stability.

In this work, we focus on the challenging task of *de novo* peptide sequencing as a **test bed** for evaluating our framework. Peptide sequencing from tandem mass spectrometry (MS/MS) data is pivotal in proteomics [2], impacting research from fundamental biology to drug development [1, 23]. However, traditional database search methods [8, 25, 45, 6, 43] falter with novel sequences, as seen in *de novo* antibody characterization [3], neoantigen discovery [17], and metaproteomics [13]. *De novo* sequencing, inferring sequences directly from spectra, is thus indispensable. The problem is well-structured, computationally demanding, and highly consequential for protein sequence modeling, making it an ideal benchmark for exploring hybrid AR–NAR architectures.

On extensive 9-species benchmarks [31, 40, 41, 49, 42], CROSSNOVO substantially surpasses AT and NAT baselines by uniquely harmonizing AT generative stability with NAT contextual awareness, yielding robust, superior performance. Ablation studies validate our architectural innovations, including the cross-decoder attention. This work significantly advances biological sequence modeling and

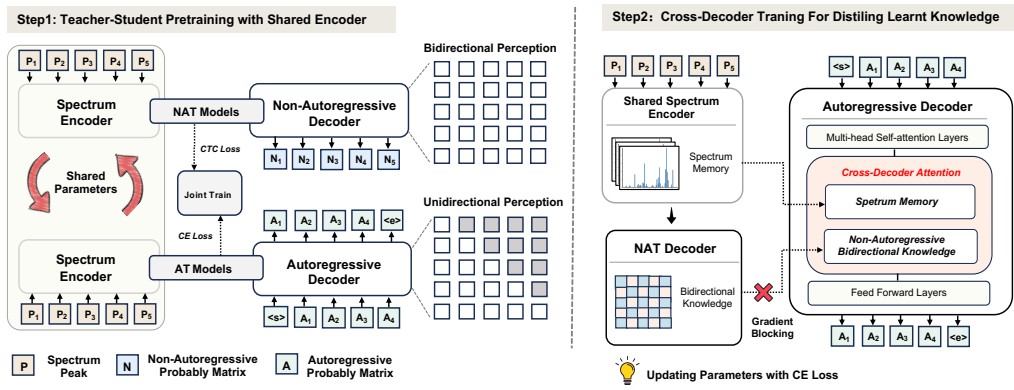

Figure 2: **The architecture of CROSSNOVO**. Step 1 involves joint training with a shared encoder in a multitask learning framework, enabling the simultaneous training of Autoregressive (AT) and Non-Autoregressive (NAT) decoders. This approach exploits the synergistic advantages of multitask learning to enhance performance. Upon convergence of both decoders, Step 2 introduces a novel knowledge distillation process, transferring insights from the NAT module to the AT module through a cross-decoder attention mechanism. Cross-decoder gradient blocking is employed throughout to optimize the training process.

*de novo* peptide sequencing, offering a novel paradigm for augmenting AT models with bidirectional understanding for complex sequence generational tasks.

## 2   Related Work

**Autoregressive and Non-Autoregressive Transformers.** Transformer architectures [32] underpin modern sequence modeling. Autoregressive (AT) variants generate tokens sequentially, ensuring high quality but suffering from inference latency [28, 42, 34]. Non-Autoregressive Transformers (NATs) [12, 21, 38] generate tokens in parallel for speed, though sometimes trading off accuracy [46, 47, 16]. In *de novo* peptide sequencing, prevalent AT models [40, 15] can suffer from error propagation. Our work pioneers a novel approach where a non-autoregressive decoder's insights into bidirectional context are used not for direct generation, but to substantially enrich and guide a primary autoregressive decoder, thereby addressing this limitation.

**De Novo Peptide Sequencing.** Early *de novo* methods [20, 9] have largely been superseded by deep learning. DEEPNOVO [31] initiated this shift using CNN-LSTMs. Transformer models now define the state-of-the-art. AT systems like Casanovo [40, 41] and its derivatives (e.g., AdaNovo [36], HelixNovo [39], InstaNovo [7], SearchNovo [37], ContraNovo [15], RankNovo [28]) have focused on architectural refinements within the AT paradigm. PrimeNovo [46], RefineNovo [48, 47], and XuanjiNovo [16] explored NAT generation for speed, though NATs can present challenges with sequence length prediction and complex optimization [19].

**CROSSNOVO** introduces a distinct and novel framework that transcends the conventional AT/NAT dichotomy. We propose a unique autoregressive architecture fundamentally enhanced by rich bidirectional latent representations. These representations are innovatively distilled from an auxiliary non-autoregressive process, allowing our model to integrate the precision of AT decoding with the holistic contextual understanding of NATs. This pioneering hybrid formulation achieves superior accuracy by uniquely leveraging the strengths of both approaches for *de novo* peptide sequencing.

## 3   Method

This section elucidates the architectural design and training paradigm of **CROSSNOVO**, a novel hybrid framework engineered for *de novo* peptide sequencing. We first establish the problem formulation and

notation. Subsequently, we dissect the core architectural innovations of **CROSSNOVO**, comprising a shared spectral feature extractor, distinct autoregressive (AT) and non-autoregressive (NAT) peptide decoders, and crucially, the sophisticated mechanisms architected for their synergistic knowledge integration.

## 3.1 Problem Formulation and Notation Overview

The core challenge of *de novo* peptide sequencing is the direct inference of an amino acid sequence $\mathbf{A} = (a_1, a_2, \ldots, a_n)$ from an observed mass spectrum $\mathcal{S}$. We formally define the input $\mathcal{S}$ as a composite of three critical information sources: (i) the peak list $\mathcal{I} = \{(mz_j, g_j)\}_{j=1}^k$, representing detected mass-to-charge ratios (mz) and their respective intensities (g); (ii) the precursor mass $\mathbf{m} \in \mathbb{R}^+$; and (iii) the precursor charge state $\mathbf{c} \in \mathbb{Z}^+$. Our objective is to learn a mapping $f : \mathcal{S} \mapsto \mathbf{A}$ that accurately predicts the ground-truth sequence.

## 3.2 Model Backbone of CROSSNOVO

The architectural blueprint of **CROSSNOVO** centers on a novel tripartite structure: a foundational shared spectrum encoder coupled with two specialized decoders—an autoregressive (AT) head for high-fidelity sequential generation and a non-autoregressive (NAT) head engineered to capture global sequence context. This dual-decoder system is designed to synergistically harness the complementary strengths of autoregressive precision and non-autoregressive holistic understanding.

**Shared Spectrum Encoder.** The core of **CROSSNOVO**'s representation learning lies in its shared spectrum encoder, a Transformer-based architecture tasked with transforming raw spectral data $\mathcal{S}$ into a rich, latent representation. We conceptualize the input peak list $\mathcal{I}$ as a sequence of tokens. For each peak $(mz_i, g_i)$, its constituent mass-to-charge ratio $mz_i$ and intensity $g_i$ are independently projected into $d$-dimensional embeddings using a domain-adapted sinusoidal encoding.

$$\mathbf{e}_j^0(\mathbf{v}) = \begin{cases} \sin((\mathbf{v})/(C \cdot (\frac{(\mathbf{v})_{\min}}{2\pi})^{\frac{2j}{d}})), & \text{for } j \leq \frac{d}{2} \\ \cos((\mathbf{v})/(C \cdot (\frac{(\mathbf{v})_{\min}}{2\pi})^{\frac{2j}{d}})), & \text{otherwise} \end{cases} \tag{1}$$

where $\mathbf{v}$ is the float value (either $mz_i$ or $g_i$) to be encoded, $C = (\mathbf{v})_{\max}/(\mathbf{v})_{\min}$ is a scaling factor, with $(\mathbf{v})_{\max}$ and $(\mathbf{v})_{\min}$ being the pre-defined bounds for the values. These per-value embeddings are subsequently fused, typically via summation, to yield an initial peak embedding sequence $\mathbf{E}^{(0)} = (\mathbf{e}_1^{(0)}, \ldots, \mathbf{e}_k^{(0)})$. This sequence is then refined through a stack of $b$ canonical self-attention layers:

$$\mathbf{E}^{(i)} = \text{SelfAttentionLayer}(\mathbf{E}^{(i-1)}) \quad \text{for } i = 1, \ldots, b. \tag{2}$$

The resultant high-level feature sequence $\mathbf{E}^{(b)} = (\mathbf{e}_1^{(b)}, \mathbf{e}_2^{(b)}, \ldots, \mathbf{e}_k^{(b)})$ constitutes the pivotal shared spectral context, concurrently utilized by both the AT and NAT decoders.

**AT Peptide Decoder.** For sequential peptide construction, **CROSSNOVO** incorporates an autoregressive (AT) decoder, which instantiates a Transformer decoder architecture carefully tailored for the nuances of peptide sequence generation. Conditioned on previously generated amino acids (or ground-truth tokens during teacher forcing), represented as embeddings $\mathbf{H}^{(0)}$, the AT decoder iteratively predicts the next amino acid $a_t$. This process unfolds over $L$ decoder layers, where each layer performs causal self-attention, producing $\mathbf{h}_t'^{(l)} = \text{CausalSelfAttn}(\mathbf{h}_t^{(l-1)}, \{\mathbf{h}_{1:t-1}^{(l-1)}\})$, to maintain autoregressive integrity. This is followed by cross-attention to the shared spectrum features $\mathbf{E}^{(b)}$, yielding $\mathbf{h}_t''^{(l)} = \text{CrossAttn}(\mathbf{h}_t'^{(l)}, \mathbf{E}^{(b)})$, and finally a feed-forward network. The output representation $\mathbf{h}_t^{(L)}$ for each token $t$ from the terminal layer sequence $\mathbf{H}^{(L)}$ is then mapped to a probability distribution over the amino acid vocabulary using $P_t(\cdot \mid \mathcal{S}, a_{<t}) = \text{softmax}(\mathcal{W}\mathbf{h}_t^{(L)})$. The token $a_t$ is then sampled from this distribution. To further bolster its predictive accuracy, and drawing inspiration from established biochemical constraints [15], our AT decoder is uniquely augmented with prefix and suffix mass information at each generation step, thereby injecting vital biological context directly into the decoding process.

**NAT Peptide Decoder.** Complementing the AT decoder, **CROSSNOVO** integrates a non-autoregressive (NAT) decoder specifically engineered to learn a holistic, global understanding of the peptide sequence from the spectrum. Diverging from the sequential nature of the AT component,

the NAT decoder generates the entire peptide sequence in a single pass. Architecturally, it mirrors a Transformer decoder but strategically omits the causal mask in its self-attention mechanisms, computing $\mathbf{v}_t'^{(l)} = \text{SelfAttn}(\mathbf{v}_t^{(l-1)}, \mathbf{V}^{(l-1)})$ to enable true bidirectional context aggregation. Adopting a common strategy for NAT models [46], we operate with a predefined maximum sequence length $T_{\max}$. A pivotal design choice for fostering effective knowledge distillation to the AT decoder (detailed in Section 3.4) is the NAT decoder's input: it exclusively receives positional embeddings $\mathbf{P}^{(0)} = (\mathbf{p}_1, \ldots, \mathbf{p}_{T_{\max}})$, without any target sequence token information. These embeddings are processed through $L'$ layers, each comprising this non-causal self-attention and subsequent cross-attention to the shared spectral features $\mathbf{E}^{(b)}$, given by $\mathbf{v}_t''^{(l)} = \text{CrossAttn}(\mathbf{v}_t'^{(l)}, \mathbf{E}^{(b)})$. The resulting latent representations $\mathbf{V}^{(L')}$ from the final NAT layer are then linearly transformed to yield predictions for all $T_{\max}$ positions concurrently.

### 3.3 Novel Joint Training Strategy for CROSSNOVO

A central contribution of **CROSSNOVO** lies in its sophisticated joint training strategy, meticulously designed to concurrently optimize both the AT and NAT decoders while fostering a unique unidirectional knowledge flow from the NAT to the AT component. This co-training paradigm leverages a shared encoder and optimizes the entire architecture via a carefully constructed multitask loss function.

**Multitask Learning Loss.** The combined loss function $\mathcal{L}_{\texttt{total}}$ is a weighted sum of the AT and NAT decoder losses:

$$\mathcal{L}_{\texttt{total}} = \lambda_{\texttt{AT}}\mathcal{L}_{\texttt{AT}} + (1 - \lambda_{\texttt{AT}})\mathcal{L}_{\texttt{NAT}} \tag{3}$$

The autoregressive loss $\mathcal{L}_{\texttt{AT}}$ is the standard cross-entropy loss:

$$\mathcal{L}_{\texttt{AT}} = -\log P(\mathbf{A} \mid \mathcal{S}; \theta) = -\sum_{t=1}^{n} \log p(\mathbf{a}_t \mid \mathbf{a}_{<t}, \mathcal{S}; \theta) \tag{4}$$

where $\theta$ represents the model parameters.

For the NAT decoder, we employ the Connectionist Temporal Classification (CTC) loss [10] to effectively handle variable-length sequences and learn robust bidirectional representations. Given a maximum generation length $T_{\max}$, the NAT decoder produces a sequence $\mathbf{y} = (\mathbf{y}_1, \ldots, \mathbf{y}_{T_{\max}})$. The CTC mechanism introduces a blank token $\epsilon$ and defines a reduction function $\Gamma$ that removes repeated tokens and then blank tokens (e.g., $\Gamma(\text{AAT}\epsilon\text{TG}) = \text{ATTG}$). The NAT loss aims to maximize the sum of probabilities of all paths $\mathbf{y}$ that can be reduced to the target sequence $\mathbf{A}$:

$$\mathcal{L}_{\texttt{NAT}} = -\log \left( \sum_{\mathbf{y}:\Gamma(\mathbf{y})=\mathbf{A}} \prod_{j=1}^{T_{\max}} P(\mathbf{y}_j | \mathcal{S}, \theta) \right) \tag{5}$$

***Importance Annealing* for Balanced Optimization.** Recognizing that the NAT decoder's strength lies in learning rich contextual priors while the AT decoder excels at fine-grained sequential prediction, we introduce a scheduling mechanism termed *importance annealing* for the weighting coefficient $\lambda_{\texttt{AT}}$. This technique, drawing conceptual parallels with curriculum learning strategies, dynamically modulates the relative contributions of the AT and NAT losses throughout training. Here, the weighting coefficient $\lambda_{\texttt{AT}}(i)$ progressively increases with training iteration $i$ according to the relation $\lambda_{\texttt{AT}}(i) = i/T_{\text{total}}$, where $i$ is the current training iteration and $T_{\text{total}}$ is the total number of iterations. This strategic annealing ensures that the NAT decoder initially plays a dominant role in shaping robust spectral representations, with the optimization landscape gradually transitioning to prioritize the AT decoder's objective for refining precise, high-fidelity sequence generation in later stages.

### 3.4 Novel Cross-Decoder Attention for Knowledge Transfer

The principal innovation enabling synergistic AT-NAT interaction within **CROSSNOVO** is our proposed *cross-decoder attention* mechanism. This novel module facilitates the direct injection of rich, bidirectionally-informed contextual knowledge, pre-learned by the NAT decoder, into the AT decoder's generation process. This transfer is typically activated during a dedicated second fine-tuning stage or at inference. Specifically, we re-engineer the AT decoder's cross-attention submodule:

**Algorithm 1 CROSSNOVO**: AT Fine-tuning with Cross-Decoder NAT Knowledge Transfer

---

1: **Inputs:** Dataset $\mathcal{D}$; Pre-trained parameters $(\theta_{\text{enc}}, \theta_{\text{AT}}, \theta_{\text{NAT}})$ from Stage 1; Fine-tuning epochs $E_{\text{ft}}$; Learning rate $\eta_{\text{ft}}$; Max NAT length $T_{\max}$.
2: **if** $E_{\text{ft}} > 0$ **then**
3:     Freeze parameters $\theta_{\text{enc}}$ and $\theta_{\text{NAT}}$.
4:     **for** epoch $e = 1$ **to** $E_{\text{ft}}$ **do**
5:         **for** each $(\mathcal{S}, \mathbf{A})$ in $\mathcal{D}$ **do**
6:             $\mathbf{E}^{(b)} \leftarrow \text{Encoder}(\mathcal{S}; \theta_{\text{enc}})$                          ▷ Use frozen encoder
7:             $\mathbf{V}^{(L')} \leftarrow \text{NATForward}(\text{PositionalEmbeddings}(T_{\max}), \mathbf{E}^{(b)}; \theta_{\text{NAT}})$    ▷ NAT features from frozen NAT decoder
8:             $\mathbf{V}^{(L')}_{\text{blocked}} \leftarrow \mathbb{GB}(\mathbf{V}^{(L')})$                ▷ Cross-decoder gradient blocking
9:             $\mathbf{C}_{\text{aug}} \leftarrow [\mathbf{V}^{(L')}_{\text{blocked}} \oplus \mathbf{E}^{(b)}]$   ▷ Augmented context for AT, cf. Eq. 8, 10 (with distinct positional encodings)
10:            $\mathcal{L}_{\text{AT-ft}} \leftarrow \text{ComputeATLossAugmented}(\mathbf{A}, \mathbf{C}_{\text{aug}}; \theta_{\text{AT}})$
11:            Update $\theta_{\text{AT}}$ using $\nabla_{\theta_{\text{AT}}} \mathcal{L}_{\text{AT-ft}}$ with $\eta_{\text{ft}}$.
12: **Return** Fine-tuned parameters $\theta_{\text{AT}}$ (and unchanged $\theta_{\text{enc}}, \theta_{\text{NAT}}$).

---

instead of solely attending to the primary spectrum features $\mathbf{E}^{(b)}$, the AT decoder's query states $\mathbf{h}_t^{\prime(l)}$ now attend to an augmented context. This augmented context is formed by concatenating the NAT decoder's final layer latent representations $\mathbf{V}^{(L')}_{\text{p}\{1:T_{\max}\}}$ with the original spectrum features $\mathbf{E}^{(b)}_{\text{p}\{T_{\max}+1:T_{\max}+k\}}$, as shown in Equation 6.

$$\mathbf{h}_t^{\text{update}} = \text{CrossAttn}\left(\mathbf{h}_t^{\prime(l)}, \left[\mathbf{V}^{(L')}_{\text{p}\{1:T_{\max}\}} \oplus \mathbf{E}^{(b)}_{\text{p}\{T_{\max}+1:T_{\max}+k\}}\right]\right) \tag{6}$$

Here, $\oplus$ denotes concatenation along the sequence dimension. The distinct positional encodings (denoted by subscript $\text{p}\{\cdot\}$) ensure the AT decoder can differentiate and appropriately leverage these heterogeneous information sources. Optimization during this knowledge transfer phase is driven exclusively by the AT loss, $\mathcal{L}_{\text{AT}}$.

***Cross-Decoder Gradient Blocking* for Stable Learning.** A critical consideration when fusing representations from independently (or jointly but distinctly) optimized modules is the potential for undesirable gradient interference. If the AT decoder's loss $\mathcal{L}_{\text{AT}}$ were allowed to backpropagate through the NAT-derived features $\mathbf{V}^{(L')}_{\text{p}\{1:T_{\max}\}}$, it could perturb the carefully learned representations optimized under the NAT-specific CTC loss ($\mathcal{L}_{\text{NAT}}$), a phenomenon we term 'representational drift.' To preempt this, we institute a *cross-decoder gradient blocking* strategy. As formalized in Equation 7, the NAT features $\mathbf{V}^{(L')}_{\text{p}\{1:T_{\max}\}}$ are treated as immutable, pre-computed contextual inputs by the AT decoder, effectively detaching them from the AT decoder's computation graph with respect to gradient flow back into the NAT decoder (e.g., via 'torch.Tensor.detach()').

$$\mathbf{h}_t^{\text{update}} = \text{CrossAttn}\left(\mathbf{h}_t^{\prime(l)}, \left[\mathbb{GB}\left(\mathbf{V}^{(L')}_{\text{p}\{1:T_{\max}\}}\right) \oplus \mathbf{E}^{(b)}_{\text{p}\{T_{\max}+1:T_{\max}+k\}}\right]\right) \tag{7}$$

where $\mathbb{GB}(\cdot)$ denotes the gradient blocking operation. This isolation ensures that the AT decoder can effectively assimilate the distilled knowledge from the NAT branch without corrupting its source, thereby promoting stable training dynamics and enhancing the performance of **CROSSNOVO**.

## 3.5 Consideration of Bidirectional Knowledge Transfer

While our proposed knowledge transfer is unidirectional (NAT→AT), one might contemplate a reciprocal AT→NAT pathway. Although architecturally plausible—for instance, by enabling the NAT decoder to attend to AT hidden states—such a design introduces a substantial risk of data contamination. Specifically, due to the teacher-forcing regimen commonly employed during AT decoder training, its internal representations are directly exposed to ground-truth target tokens. Allowing the NAT decoder to access these states would inadvertently provide it with privileged information, circumventing the fundamental challenge of *de novo* prediction from spectral data alone. Our NAT→AT transfer mechanism, in contrast, is intrinsically robust against such leakage.

The NAT decoder, by its design of solely using positional inputs for its sequence generation task (before its features are passed), learns representations purely from the spectrum and its own output distribution's constraints. Consequently, the features it provides to the AT decoder are untainted by ground-truth sequence information from the AT, ensuring the integrity of the training process. This carefully designed unidirectional distillation allows **CROSSNOVO** to effectively leverage the NAT's bidirectional context while rigorously upholding the principles of fair and challenging model training.

## 4 Experiments

### 4.1 Experiments Setup

**Dataset.** Following prior work [41, 46] for fair comparison, we trained **CROSSNOVO** on the MassIVE-KB dataset [33], which contains 30 million high-resolution peptide-spectrum matches (PSMs) from diverse instruments. For validation and benchmarking against leading methods [40, 41, 46, 50], we used the 9-species-v1 (approx. 1.5M spectra from nine experiments) and the 9-species-v2 revised datasets. The latter offers more and higher-quality spectra with broader data distribution and stricter annotation than its predecessor.

**Implementation Details.** All inputs (peaks and amino acids) were embedded into 400 dimensions. The shared spectrum encoder, NAT decoder, and AT decoder of **CROSSNOVO** each comprise 9 Transformer layers with 8 attention heads and 1024 hidden dimensions. We trained **CROSSNOVO** on eight NVIDIA A100 80GB GPUs using the AdamW optimizer [18] with an initial learning rate of $5 \times 10^{-4}$, a linear warm-up phase, and a subsequent cosine decay schedule for training stability.

**Evaluation Metrics.** Following standard practice, we evaluated performance at amino acid (AA) and peptide levels. **AA-level accuracy**: An amino acid is considered correctly predicted if its mass deviation from the actual amino acid is less than 0.1 Da, and its prefix or suffix mass differences do not exceed 0.5 Da relative to the corresponding segment of the ground truth peptide. Accuracy is then $\frac{M_{AA}}{T_{AA}}$, where $M_{AA}$ is the count of correctly predicted AAs and $T_{AA}$ is the total number of predicted AAs. **Peptide-level precision**: A peptide is considered accurately predicted if all its constituent amino acids match their true counterparts. Precision is $\frac{M_{pep}}{T_{pep}}$, where $M_{pep}$ is the count of correctly predicted peptides and $T_{pep}$ is the total number of peptides evaluated in the dataset.

**Baselines.** We benchmarked **CROSSNOVO** against a diverse set of baselines spanning three major methodological paradigms: **Database (DB)**: Represented by PEAKS [20], which performs spectral matching against a reference protein database, often using enzymatic digestion constraints (e.g., tryptic peptides) to refine the search. **Autoregressive (AT)**: This dominant paradigm includes early deep learning models like DEEPNOVO [31] and POINTNOVO [27]. More recent state-of-the-art Transformer-based AR models include Casanovo [40], HelixNovo [39], InstaNovo [7], CasanovoV2 [41] (incorporating beam search), and ContraNovo [15] (utilizing contrastive learning and amino acid mass embeddings). **Non-Autoregressive (NAT)**: A newer paradigm exemplified by PrimeNovo [46]. As the first NAR-based model for peptide generation, PrimeNovo demonstrated competitive accuracy with superior inference efficiency, highlighting the potential of NAR strategies, which **CROSSNOVO**'s hybrid architecture distinctively builds upon.

### 4.2 Results

**Performance on 9-Species-v1 Benchmark Dataset.** Table 1 highlights the superior performance of CROSSNOVO, establishing new state-of-the-art results at both amino acid and peptide levels. Our model outperforms prior autoregressive (AR) methods across all species and metrics, with amino acid recall improving from 0.785 to 0.811 and peptide recall from 0.621 to 0.654. The recall-coverage graph in Figure 1 further illustrates its dominance, consistently outperforming all baseline models at every coverage level. Moreover, our knowledge distillation techniques successfully bridge the gap between AT and NAT. Our model not only exceeds NAT in amino acid precision across all species but also surpasses NAT in peptide recall for all but two species, where it remains highly competitive. Furthermore, we observe that our combined training module has granted CROSSNOVO the advantages of both AT and NAT in predicting peptides of different species. Specifically, in Human and Mouse, where AT models performed significantly better than NAT models, CROSSNOVO extends this advantage by further outperforming NAT by 9%. In other species where NAT models performed

Table 1: Comparison of the performance of CROSSNOVO and baseline methods on the 9-species-v1 test set.

| Metrics | Architect | Methods | Mouse | Human | Yeast | M.mazei | Honeybee | Tomato | Rice bean | Bacillus | C. bacteria | Average |
|---|---|---|---|---|---|---|---|---|---|---|---|---|
| Amino Acid Precision | DB | Peaks | 0.600 | 0.639 | 0.748 | 0.673 | 0.633 | 0.728 | 0.644 | 0.719 | 0.586 | 0.663 |
| | NAT | Prime. | 0.784 | 0.729 | 0.802 | 0.801 | 0.763 | 0.815 | 0.822 | 0.846 | 0.734 | 0.788 |
| | AT | Deep. | 0.623 | 0.610 | 0.750 | 0.694 | 0.630 | 0.731 | 0.679 | 0.742 | 0.602 | 0.673 |
| | | Point. | 0.626 | 0.606 | 0.779 | 0.712 | 0.644 | 0.733 | 0.730 | 0.768 | 0.589 | 0.687 |
| | | Casa. | 0.689 | 0.586 | 0.684 | 0.679 | 0.629 | 0.721 | 0.668 | 0.749 | 0.603 | 0.667 |
| | | Insta. | 0.703 | 0.636 | 0.691 | 0.712 | 0.660 | 0.732 | 0.711 | 0.739 | 0.619 | 0.689 |
| | | Casa.V2 | 0.760 | 0.676 | 0.752 | 0.755 | 0.706 | 0.785 | 0.748 | 0.790 | 0.681 | 0.739 |
| | | Helix. | 0.765 | 0.665 | 0.768 | 0.784 | 0.757 | 0.721 | 0.793 | 0.816 | 0.681 | 0.750 |
| | | Contra. | 0.798 | 0.771 | 0.797 | 0.799 | 0.745 | 0.810 | 0.807 | 0.828 | 0.711 | 0.785 |
| | | **Ours** | **0.816** | **0.800** | **0.814** | **0.826** | **0.785** | **0.830** | **0.831** | **0.856** | **0.740** | **0.811** |
| Peptide Recall | DB | Peaks | 0.197 | 0.277 | 0.428 | 0.356 | 0.287 | 0.403 | 0.362 | 0.387 | 0.203 | 0.322 |
| | NAT | Prime. | 0.567 | 0.574 | 0.697 | 0.650 | 0.603 | 0.697 | 0.702 | 0.721 | 0.531 | 0.638 |
| | AT | Deep | 0.286 | 0.293 | 0.462 | 0.422 | 0.330 | 0.454 | 0.436 | 0.449 | 0.253 | 0.376 |
| | | Point. | 0.355 | 0.351 | 0.534 | 0.478 | 0.396 | 0.513 | 0.511 | 0.518 | 0.298 | 0.439 |
| | | Casa. | 0.426 | 0.341 | 0.490 | 0.478 | 0.406 | 0.521 | 0.506 | 0.537 | 0.330 | 0.448 |
| | | Helix. | 0.483 | 0.392 | 0.568 | 0.560 | 0.473 | 0.560 | 0.623 | 0.596 | 0.388 | 0.517 |
| | | Insta | 0.471 | 0.455 | 0.559 | 0.528 | 0.466 | 0.732 | 0.564 | 0.576 | 0.416 | 0.530 |
| | | Casa.V2 | 0.483 | 0.446 | 0.599 | 0.557 | 0.493 | 0.618 | 0.589 | 0.622 | 0.446 | 0.539 |
| | | Contra. | 0.567 | 0.622 | 0.674 | 0.630 | 0.576 | 0.672 | 0.677 | 0.688 | 0.486 | 0.621 |
| | | **Ours** | **0.596** | **0.661** | **0.698** | **0.660** | **0.610** | 0.695 | **0.716** | **0.726** | 0.518 | **0.654** |

Table 2: Comparison of the performance of CROSSNOVO and baseline methods on 9-species-v2 test set. The bold font indicates the best performance.

| Metrics | Architect | Methods | Mouse | Human | Yeast | M.mazei | Honeybee | Tomato | Rice bean | Bacillus | C.bacteria | Average |
|---|---|---|---|---|---|---|---|---|---|---|---|---|
| Amino Acid Precision | NAT | Prime. | 0.839 | 0.893 | 0.932 | 0.908 | 0.862 | 0.909 | 0.931 | 0.921 | 0.827 | 0.891 |
| | AT | Casa.V2 | 0.813 | 0.872 | 0.915 | 0.877 | 0.823 | 0.891 | 0.891 | 0.888 | 0.791 | 0.862 |
| | | Contra. | 0.839 | 0.920 | 0.919 | 0.896 | 0.848 | 0.898 | 0.913 | 0.901 | 0.807 | 0.882 |
| | | **Ours** | **0.857** | **0.937** | **0.939** | **0.920** | **0.880** | **0.914** | **0.939** | **0.927** | **0.837** | **0.906** |
| Peptide Recall | NAT | Prime. | 0.627 | 0.795 | 0.884 | 0.812 | 0.742 | **0.824** | 0.837 | 0.849 | **0.626** | 0.777 |
| | AT | Casa.V2 | 0.555 | 0.712 | 0.837 | 0.754 | 0.669 | 0.783 | 0.772 | 0.793 | 0.558 | 0.714 |
| | | Contra. | 0.616 | 0.820 | 0.854 | 0.780 | 0.711 | 0.794 | 0.799 | 0.815 | 0.575 | 0.752 |
| | | **Ours** | **0.651** | **0.850** | **0.885** | **0.819** | **0.751** | 0.816 | **0.847** | **0.850** | 0.607 | **0.786** |

better than AT models, CROSSNOVO leverages NAT strengths to increase its prediction accuracy by 1-3%.

Overall, our model's ability to integrate the strengths of both AR and NAR paradigms makes it a robust and adaptable solution. This dual capability ensures its effectiveness across diverse species, making it a valuable tool for wide-ranging biological applications.

**Performance on 9-Species-v2 Benchmark Dataset.** We evaluated **CROSSNOVO** on the 9-Species-v2 benchmark, known for its diverse modifications and high-quality spectra. As detailed in Table 2, **CROSSNOVO** achieves state-of-the-art performance, securing the highest average amino acid precision (0.906) and peptide recall (0.786) across all nine species. This demonstrates that **CROSSNOVO**'s architecture, which integrates bidirectional context into its autoregressive framework, yields substantial improvements over existing methods.

**Performance of Amino Acids with Similar Masses.** Accurately differentiating between amino acids with very similar masses is crucial for achieving precise outcomes in peptide sequencing. We conducted rigorous evaluations to assess the performance of CROSSNOVO on correctly predicting these amino acids. Specifically, our objective was to determine the efficacy of CROSSNOVO in distinguishing these challenging cases. For a comprehensive comparison, we also evaluated the performance of all AT models. The results, depicted in Figure 3, demonstrate CROSSNOVO's exceptional performance. The consistently

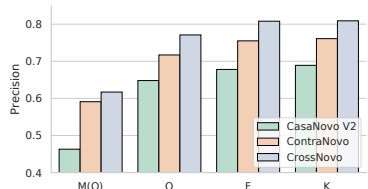

Figure 3: The precision comparison of CROSSNOVO with all AT models on amino acids with similar masses.

better performance in identifying these easily mistaken amino acids further showcases the accuracy of CROSSNOVO in a more fine-grained level.

**Sensitivity to Beam Size.** We analyzed the effect of beam size on **CROSSNOVO**'s performance using the 9-species-v1 benchmark. Table 3 shows that while larger beam sizes initially improve AA Precision and Peptide Recall,

Table 3: Effect of different beam sizes on CROSSNOVO.

| Metric | Beam Size | | | | | |
|---|---|---|---|---|---|---|
| | 1 | 3 | 5 | 7 | 9 | 11 |
| AA Precision | 0.784 | 0.804 | **0.811** | 0.810 | 0.810 | **0.811** |
| Peptide Recall | 0.634 | 0.651 | **0.654** | **0.654** | **0.654** | 0.653 |

these benefits plateau, and recall can slightly decrease with very large beams, possibly due to exposure bias [29]. A beam size of 5 offers an optimal trade-off between prediction accuracy and computational cost. Detailed results are in the Appendix (Section "Influence of Various Beam Sizes").

**Ablation Study.** The ablation study in Table 4 evaluates the effects of different proposed modules on performance. CROSSNOVO achieves its highest precision scores when both the cross decoder with gradient blocking and the shared encoder are utilized. In contrast, omitting the shared encoder while retaining the cross decoder and gradient blocking significantly reduces precision. Addi-

Table 4: Results of the ablation study showing the effects of different model configurations. The '✗' indicates training failure due to gradient explosion.

| Gradient Blocking | Cross Decoder | Shared Encoder | Amino acid Precision | Peptide Precision |
|---|---|---|---|---|
| | | ✓ | 0.795 | 0.643 |
| | ✓ | ✓ | ✗ | ✗ |
| ✓ | ✓ | | 0.698 | 0.546 |
| ✓ | ✓ | ✓ | **0.811** | **0.654** |

tionally, the absence of gradient blocking leads to training failure due to gradient explosion, as indicated by '✗'. These findings underscore the essential role of the cross-decoder with gradient blocking and the shared encoder in stabilizing and enhancing model performance.

**Downstream Task.** To demonstrate the generalizability of the proposed algorithm and its applicability, We further apply CROSSNOVO to a downstream task of identifying peptides in animal antibody data [4]. Obtaining the sequence information of antibodies is crucial for understanding the structural basis of antibody-antigen binding, recognition, and interaction [5]. However, existing methods for antibody protein sequencing rely on mRNA extraction from hybridoma cells [24], which can be challenging. De novo peptide sequencing offers a more accurate alternative by predicting peptides using MS/MS.

We utilize a publicly available human antibody dataset [30], which includes the light chain (LC) and heavy chain (HC) antibody proteins, each digested into peptides using various enzymes. We apply CROSSNOVO and several state-of-the-art baseline models to evaluate this dataset. None of these models

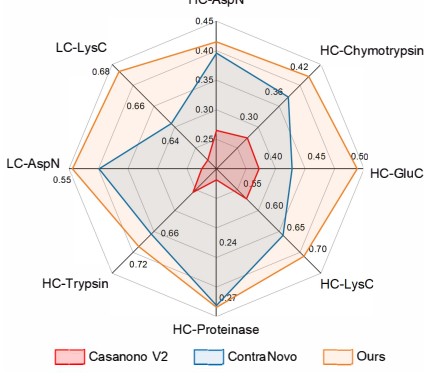

Figure 4: The peptide recall comparison of all models on Human antibody data.

were trained on antibody data, and all perform purely zero-shot inference. As shown in Figure 4, CROSSNOVO significantly outperforms the baseline models in human antibody sequencing, achieving up to a 5% improvement in both peptide recall and AA precision (Appendix Table 7). We also analyzed performance on mouse antibodies, with results detailed in the Appendix.

## 5 Conclusion

In conclusion, our research presents CROSSNOVO, a novel approach that significantly advances de novo peptide sequencing. By effectively integrating bidirectional latent knowledge from NAT to AT, CROSSNOVO leverages the strengths of both AT and NAT models. Through innovative architectural modifications, CROSSNOVO demonstrates superior performance across diverse species, surpassing both AT and NAT baselines.

## Acknowledgement

This project was fully supported by the Shanghai Artificial Intelligence Laboratory (S.S.).

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

# NeurIPS Paper Checklist

The checklist is designed to encourage best practices for responsible machine learning research, addressing issues of reproducibility, transparency, research ethics, and societal impact. Do not remove the checklist: **The papers not including the checklist will be desk rejected.** The checklist should follow the references and follow the (optional) supplemental material. The checklist does NOT count towards the page limit.

Please read the checklist guidelines carefully for information on how to answer these questions. For each question in the checklist:

- You should answer [Yes] , [No] , or [NA] .
- [NA] means either that the question is Not Applicable for that particular paper or the relevant information is Not Available.
- Please provide a short (1–2 sentence) justification right after your answer (even for NA).

**The checklist answers are an integral part of your paper submission.** They are visible to the reviewers, area chairs, senior area chairs, and ethics reviewers. You will be asked to also include it (after eventual revisions) with the final version of your paper, and its final version will be published with the paper.

The reviewers of your paper will be asked to use the checklist as one of the factors in their evaluation. While "[Yes] " is generally preferable to "[No] ", it is perfectly acceptable to answer "[No] " provided a proper justification is given (e.g., "error bars are not reported because it would be too computationally expensive" or "we were unable to find the license for the dataset we used"). In general, answering "[No] " or "[NA] " is not grounds for rejection. While the questions are phrased in a binary way, we acknowledge that the true answer is often more nuanced, so please just use your best judgment and write a justification to elaborate. All supporting evidence can appear either in the main paper or the supplemental material, provided in appendix. If you answer [Yes] to a question, in the justification please point to the section(s) where related material for the question can be found.

IMPORTANT, please:

- **Delete this instruction block, but keep the section heading "NeurIPS Paper Checklist",**
- **Keep the checklist subsection headings, questions/answers and guidelines below.**
- **Do not modify the questions and only use the provided macros for your answers**.


# A   Appendix

## A.1   Limitations

The current study validated CROSSNOVO using an extensive 9-species benchmark, which provides substantial evidence for its robust performance and generalization capabilities across these organisms. However, a focused limitation of this work is that our evaluation, while covering multiple species, did not extend to an exhaustive representation of the vast phylogenetic diversity present across all kingdoms of life. The species included in the benchmark, though diverse, may not fully encapsulate all proteomic complexities or unique peptide sequence characteristics that could be present in organisms from exceptionally divergent evolutionary lineages or those adapted to extreme and understudied environments. Consequently, while CROSSNOVO demonstrates strong cross-species performance within the considerable scope of the current benchmarks, its specific performance nuances when applied to proteomes from such highly distinct or rarely investigated species remain an avenue for potential future investigation. This would serve to further confirm the breadth of its applicability across the widest possible biological landscape.

## A.2   CTC Loss for Protein Sequence Prediction

Connectionist Temporal Classification (CTC) loss [10] is a pivotal objective function for sequence-to-sequence modeling, especially when the precise alignment between input (e.g., mass spectra $\mathcal{I}$) and output (e.g., peptide sequence $\mathcal{A}$) is unknown or variable. Our approach leverages CTC to train a model that predicts amino acid sequences from spectral data without requiring explicit alignment supervision. This section details the CTC formulation as employed in our work.

### A.2.1   Problem Definition

Given an input spectrum $\mathcal{I}$ and a target amino acid sequence $\mathcal{A} = (a_1, a_2, \ldots, a_U)$, where $U$ is the length of the target peptide, CTC aims to maximize the sum of probabilities of all possible alignment paths (or "blank-augmented" sequences) $\mathbf{z} = (z_1, z_2, \ldots, z_T)$ of length $T$ (typically the length of the model's output feature sequence) that can be reduced to $\mathcal{A}$. The reduction operation $\mathcal{B}(\mathbf{z})$ involves merging consecutive identical non-blank tokens and removing all blank tokens ($\epsilon$).

The conditional probability of the target sequence $\mathcal{A}$ given the input $\mathcal{I}$ is:

$$P(\mathcal{A}|\mathcal{I}) = \sum_{\mathbf{z}:\mathcal{B}(\mathbf{z})=\mathcal{A}} P(\mathbf{z}|\mathcal{I}). \tag{8}$$

Assuming conditional independence of outputs given the input at each time step $t$:

$$P(\mathbf{z}|\mathcal{I}) = \prod_{t=1}^{T} P(z_t|\mathcal{I})_t, \tag{9}$$

where $P(z_t|\mathcal{I})_t$ is the probability of observing token $z_t$ at time step $t$, provided by the neural network. Direct summation over all paths is intractable. Thus, we employ a dynamic programming approach.

### A.2.2   Dynamic Programming Formulation

Let $\mathcal{A}' = (\epsilon, a_1, \epsilon, a_2, \ldots, \epsilon, a_U, \epsilon)$ be the target sequence $\mathcal{A}$ interspersed with blanks at the beginning and between every token, resulting in a sequence of length $U' = 2U + 1$. We define a forward variable $\alpha_t(s)$ as the total probability of all paths that emit the prefix $\mathcal{A}'_{1:s}$ using the first $t$ time steps of the model output.

The initialization for $t = 1$ is:

$$\alpha_1(1) = P(z_1 = \epsilon|\mathcal{I})_1 \tag{10}$$
$$\alpha_1(2) = P(z_1 = a_1|\mathcal{I})_1 \tag{11}$$
$$\alpha_1(s) = 0, \quad \forall s > 2. \tag{12}$$

The recursion for $t > 1$ and $s \geq 1$ is:

$$\alpha_t(s) = (\alpha_{t-1}(s) + \alpha_{t-1}(s-1)) P(z_t = \mathcal{A}'_s|\mathcal{I})_t, \tag{13}$$

if $\mathcal{A}'_s = \epsilon$ or $\mathcal{A}'_s = \mathcal{A}'_{s-2}$ (allowing repeats of characters if separated by a blank). If $\mathcal{A}'_s \neq \epsilon$ and $\mathcal{A}'_s \neq \mathcal{A}'_{s-2}$ (i.e., $\mathcal{A}'_s$ is a distinct character that must be taken, or the previous character in $\mathcal{A}'$ was different):

$$\alpha_t(s) = (\alpha_{t-1}(s) + \alpha_{t-1}(s-1) + \alpha_{t-1}(s-2)) P(z_t = \mathcal{A}'_s|\mathcal{I})_t. \tag{14}$$

Appropriate boundary conditions (e.g., $\alpha_t(0) = 0$ for $t > 0$, and $\alpha_{t-1}(s-2) = 0$ if $s < 2$) must be handled.

The total probability $P(\mathcal{A}|\mathcal{I})$ is the sum of probabilities of paths ending in either the last blank $\mathcal{A}'_{U'}$ or the last amino acid $\mathcal{A}'_{U'-1}$ at time $T$:

$$P(\mathcal{A}|\mathcal{I}) = \alpha_T(U') + \alpha_T(U' - 1). \tag{15}$$

### A.2.3   Loss Function

The CTC loss is the negative log-likelihood of this probability:

$$\mathcal{L}_{\text{CTC}} = -\log P(\mathcal{A}|\mathcal{I}). \tag{16}$$

This formulation enables end-to-end training of our peptide sequencing model by efficiently marginalizing over all possible alignments. Numerical stability is maintained by performing computations in log-space.

## A.3   Novel Precise Mass Control (PMC) Decoding for Non-Autoregressive Transformers

We introduce Precise Mass Control (PMC), a novel knapsack-style dynamic programming algorithm specifically engineered for decoding peptide sequences from non-autoregressive transformer (NAT) models. A core challenge in applying NATs to *de novo* peptide sequencing is ensuring that generated sequences adhere to strict physical and experimental constraints. PMC addresses this by guaranteeing that the decoded peptide sequence $\mathcal{P}$ rigorously matches the experimentally observed precursor mass $m_{\text{pr}}$ within a user-defined tolerance $\delta_{\text{mass}}$. This direct integration of mass constraints into the decoding process is crucial for *de novo* sequencing, as it substantially prunes the vast search space of possible amino acid combinations, thereby significantly enhancing prediction accuracy and chemical validity. PMC offers a principled way to reconcile the parallel output generation of NATs with the precise, sequential constraints inherent in peptide mass spectrometry.

### A.3.1   Problem Formulation

Let $P_t(y|\mathcal{I})$ denote the probability distribution over the vocabulary of amino acids $\mathcal{V}_{AA}$ (augmented with a blank token $\epsilon$) for each position $t \in \{1, \ldots, T_{\text{seq}}\}$, as predicted by a NAT sequence model conditioned on input $\mathcal{I}$. Given a target precursor mass $m_{\text{pr}}$ and a mass tolerance $\delta_{\text{mass}}$, the objective of PMC is to find a peptide sequence $\mathcal{P} = (p_1, \ldots, p_L)$. This peptide $\mathcal{P}$ is derived by applying a CTC-like collapse function, $\mathcal{B}(\cdot)$, to an underlying path $\mathbf{y} = (y_1, \ldots, y_{T_{\text{seq}}})$, such that $\mathcal{P} = \mathcal{B}(\mathbf{y})$. The goal is to maximize the sum of log-probabilities of the path $\mathbf{y}$:

$$\mathbf{y}^* = \arg\max_{\mathbf{y}} \sum_{t=1}^{T_{\text{seq}}} \log P_t(y_t|\mathcal{I})$$

subject to the mass constraint:

$$m_{\text{pr}} - \delta_{\text{mass}} \leq \sum_{j=1}^{L} u(p_j) \leq m_{\text{pr}} + \delta_{\text{mass}}, \tag{17}$$

where $u(p_j)$ represents the monoisotopic mass of the amino acid $p_j$ in the peptide $\mathcal{P}$.

### A.3.2   Mass Discretization for Dynamic Programming

To facilitate the dynamic programming approach, continuous mass values are discretized. We define a discretization function $f_{\text{disc}}(\cdot)$ that maps a continuous mass to a discrete mass bin index. Consequently, the monoisotopic mass of each amino acid $aa \in \mathcal{V}_{AA}$ is mapped to its discretized counterpart, $u'(aa) = f_{\text{disc}}(u(aa))$. The target precursor mass $m_{\text{pr}}$ is also discretized to $m'_{\text{pr}} = f_{\text{disc}}(m_{\text{pr}})$, and the maximum allowable discretized mass for any peptide prefix is denoted $M'_{\text{max}}$. The tolerance $\delta_{\text{mass}}$ defines a target mass range $[m'_{\text{lower}}, m'_{\text{upper}}]$ where $m'_{\text{lower}} = f_{\text{disc}}(m_{\text{pr}} - \delta_{\text{mass}})$ and $m'_{\text{upper}} = f_{\text{disc}}(m_{\text{pr}} + \delta_{\text{mass}})$.

### A.3.3 Dynamic Programming State

The core of PMC is a DP table, $D$. An entry $D[t][m][\text{last\_y\_ne}]$ stores the maximum accumulated log-probability of a path prefix $y_1, \ldots, y_t$ such that:

1. The CTC-collapsed peptide derived from $y_1, \ldots, y_t$ has a total discretized mass $m$.

2. The last non-$\epsilon$ token encountered in the path $y_1, \ldots, y_t$ was $\text{last\_y\_ne} \in \mathcal{V}_{AA} \cup \{\text{null}\}$. The null value is used for initialization or if all preceding tokens were $\epsilon$.

Storing $\text{last\_y\_ne}$ is essential for correctly applying CTC collapse rules, particularly for handling repeats and insertions. Each entry $D[t][m][\text{last\_y\_ne}]$ stores a tuple: $(\text{log\_probability}, \text{collapsed\_peptide\_sequence})$. While we describe the top-1 (Viterbi) version for clarity, this DP formulation can be extended to beam search by storing the top-$B$ candidates in each cell.

**Initialization**  At $t = 0$, before processing any tokens: $D[0][0][\text{null}] = (0.0, \text{empty\_sequence})$. All other entries $D[0][m][\text{last\_y\_ne}]$ are initialized to $(-\infty, \text{empty\_sequence})$. The mass $m = 0$ corresponds to an empty peptide.

**Recursion**  The DP table is populated iteratively for each time step $t = 1, \ldots, T_{\text{seq}}$: For each previous discretized mass $m_{old} \in \{0, \ldots, M'_{\max}\}$: For each previous last non-$\epsilon$ token $\text{prev\_y\_ne} \in \mathcal{V}_{AA} \cup \{\text{null}\}$: If $D[t-1][m_{old}][\text{prev\_y\_ne}].\text{log\_probability} > -\infty$ (i.e., state is reachable): Let $(\text{logP}_{old}, \text{peptide}_{old}) = D[t-1][m_{old}][\text{prev\_y\_ne}]$. For each token $y_t \in \mathcal{V}_{AA} \cup \{\epsilon\}$ with probability $P_t(y_t|\mathcal{I})$: Let $\text{logP}_{cand} = \text{logP}_{old} + \log P_t(y_t|\mathcal{I})$.

1. **If $y_t = \epsilon$ (blank token):** The collapsed peptide and its mass remain unchanged. The last non-$\epsilon$ token also remains $\text{prev\_y\_ne}$.

   - $m_{new} = m_{old}$
   - $\text{peptide}_{new} = \text{peptide}_{old}$
   - $\text{next\_y\_ne} = \text{prev\_y\_ne}$

   If $\text{logP}_{cand} > D[t][m_{new}][\text{next\_y\_ne}].\text{log\_probability}$, update $D[t][m_{new}][\text{next\_y\_ne}] = (\text{logP}_{cand}, \text{peptide}_{new})$.

2. **If $y_t \in \mathcal{V}_{AA}$ and $y_t = \text{prev\_y\_ne}$ (repeat of last non-$\epsilon$ token):** This corresponds to a stutter in the $y$-sequence (e.g., $A \to AA$). The CTC-collapsed peptide and its mass do not change. The last non-$\epsilon$ token is updated to $y_t$.

   - $m_{new} = m_{old}$
   - $\text{peptide}_{new} = \text{peptide}_{old}$
   - $\text{next\_y\_ne} = y_t$

   If $\text{logP}_{cand} > D[t][m_{new}][\text{next\_y\_ne}].\text{log\_probability}$, update $D[t][m_{new}][\text{next\_y\_ne}] = (\text{logP}_{cand}, \text{peptide}_{new})$.

3. **If $y_t \in \mathcal{V}_{AA}$ and $y_t \neq \text{prev\_y\_ne}$ (new amino acid added to peptide):** The amino acid $y_t$ is appended to the collapsed peptide. Its mass is added. The last non-$\epsilon$ token becomes $y_t$.

   - $m_{new} = m_{old} + u'(y_t)$
   - $\text{peptide}_{new} = \text{peptide}_{old} \circ y_t$ (where $\circ$ denotes concatenation)
   - $\text{next\_y\_ne} = y_t$

   If $m_{new} \leq M'_{\max}$ and $\text{logP}_{cand} > D[t][m_{new}][\text{next\_y\_ne}].\text{log\_probability}$, update $D[t][m_{new}][\text{next\_y\_ne}] = (\text{logP}_{cand}, \text{peptide}_{new})$.

### A.3.4 Final Sequence Selection

After populating the DP table up to $t = T_{\text{seq}}$, the optimal peptide sequence $\mathcal{P}^*$ is determined. We iterate through all possible final non-$\epsilon$ tokens $\text{last\_y\_ne\_final} \in \mathcal{V}_{AA} \cup \{\text{null}\}$ and all discretized masses $m_{final}$ such that $m'_{\text{lower}} \leq m_{final} \leq m'_{\text{upper}}$. The peptide sequence $\mathcal{P}^*$ is the collapsed_peptide_sequence from the entry $D[T_{\text{seq}}][m_{final}][\text{last\_y\_ne\_final}]$ that has the highest log_probability among these candidates.

This PMC algorithm uniquely embeds precursor mass constraints directly within a CTC-compatible decoding framework for NATs. By ensuring adherence to experimental observations, PMC significantly refines the output of NATs for challenging tasks like *de novo* peptide sequencing, leading to more accurate and physically plausible molecular identifications.

## A.4 Influence of Various Beam Sizes

Based on the experimental results presented in Table 5, increasing the beam size generally enhances both Amino Acid Precision and Peptide Recall across various species. At both the amino acid and peptide levels, accuracy tends to improve with larger beam sizes, though it stabilizes later, exhibiting only minor increments or remaining unchanged.

For Amino Acid Precision, increasing the beam size from 1 to 3 significantly boosts average precision from 0.784 to 0.804, a rise of 0.020. Further increases up to a beam size of 11 yield smaller gains, with the highest precision of 0.811 observed at beam sizes of 5 and 11, but the rate of improvement diminishes. In terms of Peptide Recall, increasing the beam size from 1 to 3 raises average recall from 0.634 to 0.651, an increase of 0.017. Beyond a beam size of 3, improvements are marginal, with the highest recall of 0.654 achieved at beam sizes of 5, 7, and 9. Some species exhibit slight recall decreases at larger beam sizes, likely due to the Seq2Seq model's exposure bias [29, 44, 22]. The model, trained with Teacher Forcing, consistently receives the correct prior output during training but must generate its own during inference, leading to potential deviations from the optimal solution. Larger beam sizes can mitigate this issue, but excessively large sizes might cause overfitting and hinder generalization.

While larger beam sizes can enhance prediction performance, they also increase inference costs. To balance effectiveness and speed, we selected a beam size of 5 for our experiments. With this size, the model achieves high performance metrics, with an average precision of 0.811 and an average recall of 0.654, showing minimal differences compared to larger beam sizes. Additionally, compared to beam sizes of 9 or 11, a beam size of 5 offers faster inference speed and lower computational demands, maintaining prediction performance while optimizing efficiency. Therefore, considering both performance and computational costs, a beam size of 5 is considered optimal, achieving an effective balance between accuracy and efficiency.

Table 5: Comparison of Amino Acid Precision and Peptide Recall for 9-Species-v1 at Various Beam Sizes

| Metrics | Species | 1-Beam | 3-Beam | 5-Beam | 7-Beam | 9-Beam | 11-Beam |
|---|---|---|---|---|---|---|---|
| | Bacillus | 0.829 | 0.850 | 0.856 | 0.854 | 0.854 | 0.855 |
| | Clambacteria | 0.713 | 0.728 | 0.740 | 0.734 | 0.734 | 0.734 |
| | Honeybee | 0.758 | 0.779 | 0.785 | 0.785 | 0.785 | 0.786 |
| | Human | 0.766 | 0.792 | 0.800 | 0.800 | 0.802 | 0.802 |
| Amino | M.mazei | 0.801 | 0.819 | 0.826 | 0.824 | 0.825 | 0.824 |
| Acid | Mouse | 0.794 | 0.813 | 0.816 | 0.816 | 0.816 | 0.817 |
| Precision | Ricebean | 0.793 | 0.820 | 0.831 | 0.827 | 0.828 | 0.828 |
| | Tomato | 0.812 | 0.826 | 0.830 | 0.830 | 0.832 | 0.832 |
| | Yeast | 0.791 | 0.809 | 0.814 | 0.815 | 0.815 | 0.816 |
| | **Average** | 0.784 | 0.804 | **0.811** | 0.810 | 0.810 | **0.811** |
| | Bacillus | 0.706 | 0.725 | 0.726 | 0.727 | 0.726 | 0.726 |
| | Clambacteria | 0.502 | 0.517 | 0.518 | 0.519 | 0.519 | 0.518 |
| | Honeybee | 0.591 | 0.607 | 0.610 | 0.610 | 0.610 | 0.610 |
| | Human | 0.632 | 0.657 | 0.661 | 0.663 | 0.664 | 0.663 |
| Peptide | M.mazei | 0.642 | 0.658 | 0.660 | 0.660 | 0.660 | 0.660 |
| Recall | Mouse | 0.577 | 0.593 | 0.596 | 0.596 | 0.596 | 0.595 |
| | Ricebean | 0.686 | 0.712 | 0.716 | 0.717 | 0.717 | 0.717 |
| | Tomato | 0.682 | 0.694 | 0.695 | 0.694 | 0.694 | 0.694 |
| | Yeast | 0.684 | 0.696 | 0.698 | 0.698 | 0.698 | 0.698 |
| | **Average** | 0.634 | 0.651 | **0.654** | **0.654** | **0.654** | 0.653 |

### A.5 Precision-Coverage Curves

To evaluate the efficacy of our model, we utilize Precision-Coverage curves, which offer insights into performance across various species. A full visual representation of CROSSNOVO's outstanding performance is depicted in the Precision-Coverage curve shown in Figure 1. The horizontal axis represents coverage, while the vertical axis represents peptide recall. The blue line indicates the performance of Casanovo V2, the orange line represents ContraNovo, and the green line shows the performance of our model. Across all subplots, the green lines are consistently positioned above the blue and orange lines, illustrating the superior performance of our model in peptide recall over varying coverage levels. This consistent outperformance suggests potential for more accurate peptide identification, which could enhance biological insights.

## B A.7 Cross-Attention Alignment Analysis

In our extended analysis of the interaction between the Autoregressive Transformer (AT) decoder and the Non-Autoregressive Transformer (NAT) feature map, we observed a consistent and interpretable alignment pattern. Specifically, when the AT decoder generates a token at position $t$, its cross-attention weights are predominantly concentrated around the corresponding $t$-th token region in the NAT feature representation. This spatial-temporal correspondence suggests that the AT decoder has learned to selectively focus on the NAT's bidirectional contextual embedding at each generation step, rather than distributing attention uniformly across the entire sequence.

Such behavior provides strong empirical evidence that the NAT decoder functions as a powerful auxiliary encoder, supplying global contextual information that complements the AT's locally conditioned generation process. Through this mechanism, the AT decoder effectively integrates the NAT's bidirectional representation as a guiding signal, thereby enhancing both token-level consistency and long-range structural coherence in sequence generation.

### B.1 Downstream Tasks

Table 6: Comparison of the performance of CROSSNOVO and baseline methods on WIgG1-Mouse. The bold font indicates the best performance.

| Metrics | Methods | HC | | | LC | Average |
| | | AspN | Chymotrypsin | Trypsin | AspN | |
|---|---|---|---|---|---|---|
| **Amino Acid Precision** | Casa.V2 | 0.714 | 0.591 | 0.723 | 0.668 | 0.674 |
| | Contra. | 0.750 | 0.612 | 0.650 | 0.649 | 0.665 |
| | **Ours** | **0.769** | **0.640** | **0.747** | **0.724** | **0.720** |
| **Peptide Recall** | Casa.V2 | 0.557 | 0.483 | 0.636 | 0.456 | 0.533 |
| | Contra. | 0.649 | 0.545 | 0.671 | 0.519 | 0.596 |
| | **Ours** | **0.662** | **0.577** | **0.699** | **0.581** | **0.630** |

Table 7: Comparison of the performance of CROSSNOVO and baseline methods on IgG1-Human. Bold values indicate the best performance.

| Metrics | Methods | HC | | | | | | LC | | Average |
| | | AspN | Chymo. | GluC | LysC | Proteinase | Trypsin | AspN | LysC | |
|---|---|---|---|---|---|---|---|---|---|---|
| **Amino Acid Precision** | Casa.V2 | 0.520 | 0.472 | 0.605 | 0.757 | 0.354 | 0.759 | 0.666 | 0.778 | 0.642 |
| | Contra. | 0.580 | 0.565 | 0.642 | 0.790 | 0.348 | 0.787 | 0.702 | 0.793 | 0.676 |
| | **Ours** | **0.613** | **0.617** | **0.694** | **0.814** | **0.367** | **0.803** | **0.719** | **0.807** | **0.702** |
| **Peptide Recall** | Casa.V2 | 0.265 | 0.274 | 0.399 | 0.569 | 0.206 | 0.595 | 0.325 | 0.625 | 0.446 |
| | Contra. | 0.396 | 0.372 | 0.437 | 0.653 | 0.274 | 0.675 | 0.499 | 0.646 | 0.529 |
| | **Ours** | **0.415** | **0.421** | **0.512** | **0.701** | **0.275** | **0.699** | **0.544** | **0.676** | **0.560** |

**Data.** The Human IgG1 antibody dataset (IgG1-Human) consists of mass spectrometry data collected using the LTQ Orbitrap instrument. Ionization was performed through higher-energy collisional dissociation (HCD), and the resulting peptide fragments were captured at a resolution of 17,500 FWHM. The dataset features peptides digested by a variety of proteolytic enzymes, including trypsin, chymotrypsin, asp-N, lys-C, glu-C, and proteinase K. We perform the evaluation on donwloaded data with no processing.

The Mouse IgG1 antibody dataset (WIgG1-Mouse) was similarly analyzed using an LTQ Orbitrap mass spectrometer with HCD ionization and the same resolution of 17,500 FWHM. In this dataset, the mouse peptides were digested with trypsin, asp-N, and chymotrypsin to generate a comprehensive proteomic profile. We also used downloaded data for evaluation of all tested models.

**Results.** As shown in Tables 7 and 6, CrossNovo consistently achieves superior performance across both amino acid-level precision and peptide-level recall when compared to the baseline methods. Specifically, for the Mouse dataset, CrossNovo demonstrates a notable improvement in peptide recall, achieving up to a 6% increase for the AspN enzyme on the light chain (LC) protein. Similarly, in the Human dataset, the peptide recall improvement is up to 4.5% for the AspN enzyme on the LC protein.

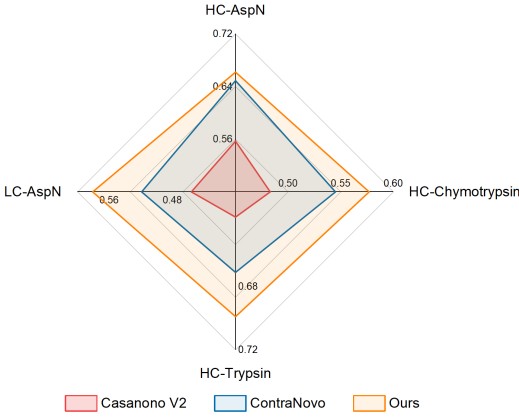

Figure 5: The comparison of performance of models in mouse antibody data.

The performance gain is particularly pronounced for light chain proteins across both species, with CrossNovo showing higher overall precision and recall. The average amino acid precision for the Mouse dataset reaches 0.720, while the peptide recall is boosted to 0.630. For the Human dataset, CrossNovo attains an average precision of 0.702, with a peptide recall of 0.560, further underscoring its effectiveness over baseline approaches.

These differences in performance are even more apparent in Figure 5, where CrossNovo's enhancements, particularly on the light chain proteins, can be clearly visualized. The consistency in performance improvements across both datasets highlights CrossNovo's ability to handle diverse proteolytic enzymes with high accuracy, especially in cases involving the light chains.

### B.2 Post-Translational Modification (PTM) Fine-Tuning Evaluation

To further evaluate the generalization ability of CROSSNOVO and PRIMENOVO, we extend our experiments to post-translational modification (PTM) prediction tasks. Each PTM model was fine-tuned independently on the curated **21-PTM dataset**, containing spectra annotated with 21 types of peptide modifications. For conciseness, we report results on the first five PTMs in alphabetical order: Acetylation, Biotinylation, Crotonylation, Butyrylation, and Dimethylation.

**Analysis.** As shown in Table, both models maintain high classification accuracy after PTM-specific fine-tuning. CROSSNOVO consistently achieves superior or comparable results across all metrics, particularly in amino acid and peptide recall for complex modifications like Butyrylation and Dimethylation. This demonstrates the advantage of cross-decoder attention and bidirectional contextual transfer in recognizing subtle mass shifts introduced by PTMs. Furthermore, fine-tuning on the

Table 8: Performance comparison on five representative PTMs after fine-tuning on the 21-PTM dataset. Bold values indicate the best performance for each column.

| Metric | Model | Acetylation | Biotinylation | Crotonylation | Butyrylation | Dimethylation |
|---|---|---|---|---|---|---|
| Classification Accuracy | PrimeNovo | 0.98 | **0.99** | 0.98 | 0.97 | 0.95 |
| | CrossNovo | **0.98** | 0.98 | **0.98** | **0.98** | **0.97** |
| AA Recall | PrimeNovo | 0.95 | **0.83** | **0.90** | 0.82 | 0.84 |
| | CrossNovo | **0.96** | **0.83** | 0.86 | **0.85** | **0.85** |
| Peptide Recall | PrimeNovo | 0.89 | 0.70 | **0.79** | 0.65 | 0.66 |
| | CrossNovo | **0.90** | **0.72** | 0.78 | **0.69** | **0.68** |

diverse 21-PTM dataset highlights the scalability of our architecture to biochemical modifications, extending its applicability beyond canonical peptide sequencing.

## B.3 Broader Impact

Advancements in *de novo* peptide sequencing have far-reaching societal implications, particularly in healthcare, biotechnology, and life sciences. CROSSNOVO significantly enhances the accuracy and robustness of sequence prediction, directly benefiting applications such as neoantigen discovery for personalized cancer immunotherapy, rapid characterization of therapeutic antibodies, and profiling of microbial communities in environmental and clinical metaproteomics. By improving the capacity to identify novel or mutated peptides without reliance on existing databases, our method contributes to overcoming current bottlenecks in drug discovery, vaccine development, and disease diagnostics. Moreover, CROSSNOVO's hybrid architecture—balancing predictive stability with rich contextual understanding—enables more reliable deployment in translational pipelines where interpretability and robustness are critical. However, as with any powerful bioinformatic tool, careful consideration must be given to issues of data privacy, equitable access to computational resources, and responsible clinical translation to ensure these technologies are used ethically and inclusively.

