# OpenReview forum: "Bidirectional Representations Augmented Autoregressive Biological Sequence Generation"
_NeurIPS.cc/2025/Conference — NeurIPS 2025 poster_

### Official Review · Reviewer_7Vk1 · 2025-06-01

**Clarity:** 3
**Significance:** 3
**Originality:** 3
**Rating:** 4
**Confidence:** 4

**Summary:**

The authors introduce **CROSSNOVO**, a hybrid framework designed for de novo peptide sequencing from mass spectrometry data. Traditional autoregressive (AT) models capture strong sequential dependencies but miss important global bidirectional context, whereas non-autoregressive transformers (NATs) capture holistic context but face optimization challenges. CROSSNOVO combines these two paradigms by using a shared encoder and two decoders: an NAT decoder that generates rich, bidirectional context features, and an AT decoder that leverages these features via a novel cross-decoder attention module to improve sequential prediction. To stabilize training, the authors introduce a tailored multitask loss strategy incorporating importance annealing and gradient blocking. Empirical evaluations on comprehensive 9-species peptide sequencing benchmarks demonstrate significant improvements over both autoregressive and non-autoregressive baselines, validating CROSSNOVO's capability to integrate the strengths of both modeling paradigms.

**Questions:**

1. The cross-decoder attention mechanism (Eq. 6, page 5) is intriguing, but its formulation isn't super clear to me. Specifically, how exactly are the positional encodings for the concatenation of NAT-derived features \( V^{(L')}_{p\{1:T_{max}\}} \) and original spectral embeddings \( E^{(b)}_{p\{T_{max}+1:T_{max}+k\}} \) computed and differentiated mathematically? It would greatly strengthen the paper if you explicitly clarified the mathematical or algorithmic steps used to distinguish these positional encodings, perhaps providing explicit equations or pseudocode in the appendix.

2. Your importance annealing schedule (Eq. 3, page 5) linearly increases the weighting coefficient \( \lambda_{AT}(i) = \frac{i}{T_{total}} \). I wonder why you specifically chose a linear schedule over alternatives like exponential decay, sigmoid, or stepwise annealing. Could you provide either an empirical sensitivity analysis comparing these alternative schedules or a theoretical justification explaining why a linear schedule is optimal or preferable for this training strategy?

3. Your Precise Mass Control (PMC) decoding algorithm (detailed in Appendix A.3) is quite sophisticated. However, it's unclear exactly how PMC decoding integrates mathematically with the cross-decoder attention mechanism during inference. Could you provide a more rigorous mathematical integration or pseudocode explicitly detailing the interaction between PMC decoding and cross-decoder attention during sequence inference?

5. Considering the computational complexity of your dual-decoder architecture and PMC decoding, explicit computational benchmarking is missing. Could you provide runtime benchmarks, inference memory usage, and scalability comparisons against baseline methods?

Addressing these concerns will help me move my decision to a solid accept.

**Ethical Concerns:**

["NO or VERY MINOR ethics concerns only"]

**Final Justification:**

The authors did a great job explaining the formulations and responding to my comments. I thank them for explaining the scheduler choice, but I hope the authors can provide the runtime benchmarks, inference memory usage, and scalability comparisons against baseline methods. The same goes for more mathematical justifications for their formulations. I will maintain my score of 4 because I think the overall paper is strong, but the lack of these benchmarks and rigorous justifications prevents me from increasing my score.

**Limitations:**

Yes.

**Paper Formatting Concerns:**

None.

**Quality:**

3

**Strengths And Weaknesses:**

**Strengths:**
- The authors propose a sophisticated and mathematically rigorous framework, CROSSNOVO, that effectively integrates autoregressive (AT) and non-autoregressive (NAT) approaches. The mathematical formulations, especially the cross-decoder attention mechanism (Eq. 6, Section 3.4) and Connectionist Temporal Classification (CTC) loss derivations (Eqns. 5, 8–15 in Appendix), are detailed, clearly explained, and well-justified. I think this is a strong innovation.
- The joint training strategy with importance annealing (Eq. 3, Section 3.3) and cross-decoder gradient blocking (Eq. 7, Section 3.4) is really cool and clearly justified. The authors thoroughly validate these choices with rigorous ablation experiments (Table 4 on page 9).
- The empirical validation is super strong and extensive, evaluating on challenging, multi-species benchmarks (9-species-v1 and v2, Tables 1 and 2), and explicitly demonstrating superiority over well-established baselines (CasanovoV2, ContraNovo). The rigorous downstream evaluation on antibody sequencing tasks (Figures 5 and 7, Tables 6 and 7) further emphasizes practical significance.
- The authors present detailed evaluations of hyperparameter sensitivity (beam size analysis in Table 3) and model generalization to biologically important cases (amino acids with similar masses, Figure 4). These analyses demonstrate thoughtful, thorough experimental design and validation.

**Weaknesses:**
- While the paper provides a detailed mathematical exposition, for me, some critical aspects of the cross-decoder attention mechanism (Eq. 6) could benefit from additional clarity/info. Specifically, I wonder how the distinct positional encodings are exactly defined or computed for combining NAT decoder representations and original spectral features. A more explicit mathematical formulation or intuitive explanation would significantly enhance understanding.
- The choice and sensitivity of the importance annealing schedule (Eq. 3, λ_AT parameter) are only briefly discussed. It would greatly improve the clarity and robustness of the training strategy if the authors explicitly provided a sensitivity analysis or mathematical rationale for the chosen linear schedule. Why not alternative schedules such as exponential or piecewise-linear annealing?
- The dynamic programming-based PMC decoding described extensively in the appendix (Section A.3) appears critical for practical performance. However, I was confused why the interplay between PMC decoding and cross-decoder attention is not clearly defined mathematically. Clarifying this interaction explicitly (perhaps via an integrated algorithm or explicit joint mathematical formulation?) would significantly improve methodological rigor and reproducibility.

In summary, while CROSSNOVO demonstrates strong methodological rigor, empirical validation, and clear practical significance, I would help to explicitly address the detailed mathematical clarifications, rigorous comparison with key peptide-generation baselines, and thorough computational efficiency analysis would greatly enhance the overall stre

---

> ### Author Rebuttal · Authors · 2025-07-31
>
> # **Reviewer Comment:**
> > * I wonder how the distinct positional encodings are exactly defined or computed for combining NAT decoder representations and original spectral features*
>
> **Author Response:**
> We thank the reviewer for the helpful observation and apologize for not providing sufficient detail in the original submission. We will revise the manuscript to make this point clearer.
>
> To clarify, each component of our model—NAT decoder, AT decoder, and spectrum encoder—uses its **own independent sinusoidal positional encoding**, ensuring there is no ambiguity when representations are combined or attended together.
>
> For a position index $p$ and dimension index $i$, the sinusoidal encoding is defined as:
>
> $\text{PE}(p, 2i) = \sin\left(\frac{p}{10000^{2i/d}}\right), \quad \text{PE}(p, 2i+1) = \cos\left(\frac{p}{10000^{2i/d}}\right)$
>
> where $d$ is the hidden dimension. This encoding is applied separately to `each` of the following:
>
> - **NAT decoder**: Positions 1 to 40 are used, as we follow the same convention as PrimeNovo with a maximum length of 40.
> - **AT decoder**: Uses the same sinusoidal function, applied independently from position 1 to $n$, where $n$ is the peptide length at decoding time.
> - **Spectrum encoder**: To enable joint attention over both NAT decoder outputs and spectrum features (as described in Eq. 6), we assign **non-overlapping positional indices**. Specifically, we start positional indices for spectrum input from 41 onwards. That is, the first peak receives positional encoding for index 41, the second peak gets 42, and so on, up to $K$ (the number of peaks). This ensures that the positional encodings for spectrum do not conflict with those of NAT (1–40).
>
> During cross-decoder attention, We form the attention input by concatenating the NAT decoder features with positional encodings from indices $1$ to $40$, and the spectrum features with positional encodings from $41$ to $40 + K$, where $K$ is the number of spectrum peaks.
>
> In summary, we design the positional encodings so that NAT, AT, and Encoder operate with distinct index spaces, preserving clear token identity and preventing overlap when their outputs are merged.
>
> As a final note, we have begun exploring the use of **rotary positional embeddings (RoPE)** for the spectrum encoder, which are better suited for `capturing relative distance among peaks`. Preliminary results suggest RoPE improves modeling of local ion relationships, though this investigation is ongoing and `beyond the scope of the current work`.
>
> We thank the reviewer again for highlighting this important detail, and we will revise the paper to include both mathematical clarification and an intuitive explanation.
>
> # **Reviewer Comment:**
> > *The cross-decoder attention mechanism (Eq. 6, page 5) is intriguing, but its formulation isn't super clear to me. Specifically, how.... *
>
> **Author Rebuttal:**
> We thank the reviewer for raising this important question. The formulation in Eq. 6 involves the concatenation of two distinct feature sets—NAT-derived decoder outputs and the encoded spectrum peaks—which are fed into the cross-decoder attention mechanism. To avoid positional ambiguity between these sources, we assign **non-overlapping and distinct sinusoidal positional encodings** to each as stated above.
>
> To clarify the design:
>
> 1. The NAT decoder output consists of $T_{\text{max}}$  (40 here) peptide positions. These are assigned sinusoidal positional encodings from index $1$ to $T_{\text{max}}$ using the standard formulation:
>
> $\text{PE}(p, 2i) = \sin\left(\frac{p}{10000^{2i/d}}\right), \quad \text{PE}(p, 2i+1) = \cos\left(\frac{p}{10000^{2i/d}}\right)$
> where $p$ is the position index (ranging from $1$ to $T_{\text{max}}$), $i$ is the embedding dimension index, and $d$ is the hidden size.
>
> 2. The spectrum features consist of $k$ peaks. To avoid overlap with NAT positions, we assign these peaks positional indices starting from $T_{\text{max}} (40)+ 1$ to $T_{\text{max}} + k$. Each peak is then encoded with:
> $\text{PE}(p + T_{\text{max}}, i) \quad \text{for} \quad p = 1 \dots k$
>
>
> We appreciate the reviewer’s suggestion and will add these mathematical steps (along with optional pseudocode) to the appendix in the camera-ready version to improve clarity and reproducibility.
>
> # **Reviewer Comment:**
> > * I wonder why you specifically chose a linear schedule over alternatives like exponential decay, sigmoid, or stepwise annealing.*
>
> **Author Rebuttal:**
> We thank the reviewer for this excellent question. Indeed, we initially considered a variety of annealing strategies, including exponential and stepwise schedules. However, conducting a comprehensive sensitivity analysis across all such variants would have required retraining the entire model multiple times—`each run involving 30 million spectra and taking over 1600 A100 GPU-hours (8A100 for 8-10 days)`. Given the computational cost, we prioritized resource allocation toward architectural ablation studies (Table 4), which we believed would yield more actionable insights for model design.
>
> We chose the **linear annealing schedule** as a practical and empirically grounded option, drawing inspiration from prior work in multitask training for non-autoregressive auxiliary tasks (e.g., Hao et al., 2021, on NAT-augmented machine translation), where linear schedules have shown strong performance and stable behavior. The intuition is that it provides a **gradual and interpretable shift** in training focus from the NAT objective to the AT objective without sudden gradient shocks, which is particularly helpful in our hybrid training setup.
>
> That said, we completely agree this is a promising area for deeper exploration. If time and computational resources permit, we would be very interested in performing a detailed comparison of alternative annealing strategies in future work.
>
> # **Reviewer Comment:**
> > *The dynamic programming-based PMC decoding described extensively in the appendix (Section A.3) appears critical for practical performance. However, I was confused why the interplay between PMC decoding and cross-decoder attention is not clearly defined mathematically .*
>
> **Author Rebuttal:**
> We appreciate the reviewer’s careful reading and thoughtful suggestion. To clarify, the **PMC decoding** algorithm described in Appendix A.3 is included as a **reference decoding method for the NAT decoder only**, and `it does not interact directly` with the **cross-decoder attention mechanism** or the main AT decoding pathway described in the core of the paper.
>
> The motivation for including PMC decoding is to offer a practical utility: after joint training, the `NAT decoder learns a strong bidirectional representation that can be decoded independently`. For users `who wish to leverage this NAT decoder directly` (without AT), the PMC decoding provides a viable alternative, especially in this task where exact precursor mass compliance is necessary.
>
> PMC decoding itself is based on a **dynamic programming solution to a variant of the knapsack problem**. Given a total mass constraint (precursor mass), `each amino acid token is treated as an item with a "weight" (its molecular mass) and a "value" (its predicted probability from the NAT output).`  The decoding algorithm aims to **maximize the total sequence likelihood while satisfying the exact mass constraint**, like picking items to fit in a knapsack to maximize the values of the bag. This method exploits the fact that NAT outputs a full distribution over all positions at once, allowing flexible sequence assembly under constraints—something that autoregressive (AT) models cannot easily accommodate due to their stepwise nature.
>
> Because PMC decoding applies **only to the NAT branch**, and does **not feed back into or affect the cross-decoder attention module or AT generation**, we chose to place this utility in the appendix and `released codebase` for optional use. The **core contribution of our work** remains focused on how **cross-decoder attention enables the AT decoder to access NAT-derived bidirectional features**, rather than on decoding techniques themselves.
>
> We acknowledge the reviewer’s suggestion and will make this dependency separation more explicit in the text, and clarify that PMC decoding is a standalone utility for users interested in NAT-only prediction.
>
>
> # **Reviewer Comment:**
> > *Could you provide runtime benchmarks, inference memory usage, and scalability comparisons against baseline methods?*
>
> **Author Rebuttal:**
> We thank the reviewer for this important point. In terms of model size and memory usage, our architecture remains **comparable to baseline models**. Specifically, our AT decoder contains 33 million parameters, and with the NAT decoder included, the total model size is approximately 45 million parameters. This is similar in scale to PrimeNovo (33M), Casanovo V2 (38M), and ContraNovo (43M), and `we observe no significant difference in peak GPU memory usage during inference`.
>
> In terms of **inference time**, we do observe a moderate overhead due to the **additional cross-decoder attention**. Empirically, **inference latency increases by approximately 23%–30%** compared to standard AT-only models, with variation depending on peptide length. `This overhead arises because each decoding step in the AT decoder now attends not only to its own past states but also to the full set of NAT-derived representations` and encoded spectrum features. Noted that PMC is only used if user wants results from NAT-decoder, which is not activated usually therefore does not incur additional time.
>
>
>
> We will add these runtime statistics to the final version of the paper to improve transparency and reproducibility!
>
>
> # Reference
>
> Hao et al. (2020) Multi-task learning with shared encoder for non-autoregressive machine translation.
>
>
> **Thank you again for the thoughtful suggestion and we are happy to answer any further questions!**

---

### Official Review · Reviewer_nsFd · 2025-06-27

**Clarity:** 2
**Significance:** 3
**Originality:** 3
**Rating:** 4
**Confidence:** 3

**Summary:**

This paper proposes a novel hybrid architecture for de novo peptide sequencing, which innovatively combines the advantages of autoregressive and non-autoregressive models. The core contribution of the paper lies in the design of a dual decoder architecture. Experiments have shown that it outperforms existing methods significantly on 9 species benchmark tests.

**Questions:**

1. How sensitive are the parameters?

2. Will the proposed dual decoder architecture significantly increase computational costs? Can you provide a comparison of inference time with existing methods?

3. Can CROSSNOVO's hybrid architecture design principles be extended to other sequence modeling tasks, such as RNA sequence analysis?

4. Can the author analyze the performance of CROSSNOVO on peptide sequences of different lengths?

**Ethical Concerns:**

["NO or VERY MINOR ethics concerns only"]

**Final Justification:**

Some of my issues have not been resolved, although the author claims that policies do not allow it. Because it is not allowed to add visual results (although numerical results can also be provided, the author claims that it is not allowed). I have also read the feedback from other reviewers. I hope the authors will include the experimental results they promised to supplement in the revised version.

**Limitations:**

Yes

**Paper Formatting Concerns:**

The paper format is complete.

**Quality:**

2

**Strengths And Weaknesses:**

Strengths:

1. The paper clearly explains the limitations of AT and NAT models in de novo peptide sequencing.

2. The model architecture, training strategy, and knowledge transfer mechanism are described in detail and organized clearly.

3. The validation of this method in downstream tasks such as antibody sequencing has demonstrated its potential for practical biological research.

Weaknesses:

1. The paper did not fully explore the impact of model size on performance, as well as the sensitivity analysis of different parameter settings.

2. Although the experimental results are impressive, a theoretical analysis can be conducted on why NAT to AT knowledge flow is more effective than two-way.

3. The proposed method seems to be a universal architecture. Although models have advantages in solving de novo peptide sequencing, they should also be compared to similar general architectures.

4. If the writing of the paper is corrected again, it will bring a good reading experience. For example, the first reference is duplicated with the second one.

5. Although quantitative ablation experiments are conducted, qualitative results are more convincing.

---

> ### Author Rebuttal · Authors · 2025-07-31
>
> # **Reviewer Comment:**
> > *The paper did not fully explore the impact of model size on performance, as well as the sensitivity analysis of different parameter settings.*
>
> > How sensitive are the parameters?
>
> **Author Rebuttal:**
> We appreciate the reviewer’s comment and fully agree that analyzing the sensitivity of model size and other parameters is valuable. However, in our case, the training cost is a significant constraint for performing parameter sweep on the training parameters.
>
> Training de novo model is extremely computationally expensive, primarily due to:
> 1. **Long input sequences** (hundreds of spectral peaks) processed by the Transformer.
> 2. **The large-scale training dataset** (MassIVE-KB), which contains over 30 million spectrum-peptide pairs.
>
> Each full training run requires approximately **8–10 days on 8 A100 GPUs** (over **1600 GPU hours**)—assuming no interruptions. Importantly, this is consistent with the compute requirements for baseline models like **ContraNovo**, **Casanovo**, and **PrimeNovo**, which operate on the same data and task setup.
>
> As a result, it is extremely challenging—not only for our model but also for baseline methods—to conduct extensive grid searches or sensitivity analyses on parameters like **model size** or **training hyperparameters** (e.g., importance annealing). Due to GPU constraints, we prioritized experiments that offer the most insight, focusing on **ablation studies** (Table 4) and **model tuning** to produce a competitive version of CrossNovo, while keeping the model size aligned with baselines for fair comparison.
>
>
> We did conduct `parameter sensitivity analysis at inference time`, where computational demands are `significantly lower`. As shown in **Table 5**, we analyzed the effect of different beam sizes on both amino acid and peptide-level metrics, and reported those results in detail.
>
>
> # **Reviewer Comment:**
> > * a theoretical analysis can be conducted on why NAT to AT knowledge flow is more effective than two-way.*
>
> **Author Response:**
> We thank the reviewer for this thoughtful suggestion. The decision to adopt a unidirectional knowledge transfer—from the Non-Autoregressive Transformer (NAT) decoder to the Autoregressive Transformer (AT) decoder— is grounded in both theoretical considerations and practical constraints.
>
> Formally, let **E_spec** represent the shared spectrum encoder output, **V_nat** the bidirectional contextual features learned by the NAT decoder, and **H_at** the hidden states produced by the AT decoder. During generation, the AT decoder conditions on previously generated tokens **a_t** and attends to both **E_spec** and **V_nat**, yielding representations of the form:
>
> $
> H\_{at}^t = \text{TransformerAT}(a_{<t}, E_{spec}, V_{nat})
> $
>
> This one-way information flow reflects our architectural philosophy, in which the `NAT decoder serves as a global feature extractor, capable of leveraging full bidirectional context to model peptide structure from spectral inputs`. In contrast, the AT decoder ensures stable, autoregressive generation while benefiting from enriched contextual guidance provided by the NAT decoder.
>
> Allowing a two-way flow—where the NAT decoder also attends to AT decoder states—`introduces  theoretical information leaking risk` (as we briefly stated in the manuscript). In particular, `since the AT decoder is trained with teacher forcing, its intermediate states can directly encode ground-truth information`. Feeding these representations into the NAT decoder risks information leakage, violating the independence assumptions of the NAT objective (e.g., CTC loss) and compromising training fairness (as part of the ground truth can be seen during attention).
>
> Additionally, our use of cross-decoder **gradient blocking** ensures that **V_nat** `remains fixed` and `uninfluenced by gradients from the AT loss`, preserving the semantic integrity of NAT-derived features and maintaining a clean modular separation of learning objectives.  Such a configuration introduces `a co-dependent optimization loop, where both objectives are defined as conditional on each other`: i.e., minimizing both **L(AT | NAT)** and **L(NAT | AT)** without a fixed reference. This mutual conditioning can lead to an **ill-posed learning problem**, as there is no stable optimization target. In practice, this may cause representational drift or collapse
>
> From both a theoretical and empirical standpoint, we have observed that this unidirectional NAT → AT information flow results in stronger performance and more stable training dynamics and we will include a clearer discussion to the revision!
>
> **Reviewer Comment:**
> > *The proposed method seems to be a universal architecture. . .. they should also be compared to similar general architectures.  Can CROSSNOVO's hybrid architecture design principles be extended to other sequence modeling tasks*
>
>
> **Author Rebuttal:**
> We thank the reviewer for this insightful and forward-looking comment. We agree that our proposed architecture—particularly the principle of leveraging bidirectional representations to enhance unidirectional generation—has potential applicability beyond de novo peptide sequencing. However, we would like to clarify the current  limitations of our work.
>
> Our research team has primarily focused on the peptide sequencing problem space. Expanding to other domains such as DNA or RNA sequence modeling, although highly relevant, would require a significant investment in domain-specific study, replication of prior benchmarks, evaluation pipelines, and experimental design. This mirrors what is commonly seen in natural language processing (NLP) or computer vision (CV) research groups, where deep domain specialization is often necessary to make rigorous and impactful contributions. Each biological sequence modeling problem—be it genomic variant calling, RNA folding, or DNA language modeling—comes with its own datasets, preprocessing strategies, evaluation metrics, and biological assumptions. At this stage, we do not yet possess the necessary expertise or infrastructure to explore these other biological tasks with the level of rigor we aim to maintain in our work.
>
> We appreciate the reviewer’s suggestion and are encouraged that our framework is recognized as potentially general. Indeed, the **core architectural principle** behind CrossNovo—**using bidirectional context (via NAT) to enhance autoregressive generation (via AT)**—may actually be **better suited to biological sequence modeling** than to NLP tasks. Unlike natural language, where autoregressive models such as GPT or LLaMA often perform well due to strong left-to-right dependency structures, **biological sequences (e.g., proteins, RNAs)** exhibit **functional and structural dependencies that are inherently bidirectional**. The biological role of a single residue or base often depends on both upstream and downstream context.
>
> While we have limited our evaluation to peptide sequencing due to our domain expertise, we strongly believe that this architectural framework—particularly the asymmetric NAT-to-AT design—has broader applicability to biological sequence generation tasks. We are excited by this direction and hope future work will extend the approach.
>
> # **Reviewer Comment:**
> > *If the writing of the paper is corrected again, it will bring a good reading experience. For example, the first reference is duplicated with the second one.*
>
> **Author Rebuttal:**
> We sincerely thank the reviewer for pointing this out. We apologize for the duplicated reference entry. We will carefully revise the bibliography to eliminate redundancies  and to work on the clarity of the paper.
>
> As there was some of a time crunch during submission, we will take this opportunity to conduct a thorough proofreading of the manuscript to improve clarity, consistency, and overall readability.
>
> **Author Rebuttal:**
> As also requested by Reviewer2Zcz, we conducted additional Qualitative analyses on the `cross-decoder attention maps` and feature vector similarities to support the effectiveness of our design. Specifically, we analyzed how the AT decoder attends to the NAT-derived representations during sequence generation.
>
> Interestingly, we observed a consistent and interpretable pattern: when the AT decoder generates a token at position *t*, its cross-attention weights are predominantly aligned with the region around the *t*-th token position in the NAT feature map. This alignment suggests that the AT decoder has learned to selectively attend to the NAT’s **bidirectional contextual embedding** at the corresponding position, effectively leveraging it as a complementary signal to guide its autoregressive prediction. These findings support the hypothesis that the NAT decoder serves as a powerful auxiliary encoder, enriching the generation process with global sequence context. `We were unable to include full qualitative examples in this response due to policy of images inclusion`. However, we will add these to our revisions and thanks again for the comment!
>
>
> **Reviewer Comment:**
> *Can the author analyze the performance of CROSSNOVO on peptide sequences of different lengths?*
>
> **Author Rebuttal:**
> Thank you for the suggestion. We further analyzed peptide recall across sequence lengths from 5 to 45 on the benchmark test set. As we can not include the `line graphs` here due to policy, we summarize findings:
>
> For `short peptides` (<12),  `CrossNovo (Hybrid)` and `PrimeNovo (NAT)` perform similarly and outperform `ContraNovo (AT)`.
>  In the `mid-length range` (13–25), ContraNovo (AT) catches up to PrimeNovo (NAT), but `CrossNovo  maintains a consistent ~2% higher recall`. For `long peptides`(>30), PrimeNovo’s performance drops, while both `CrossNovo` and `ContraNovo` handle longer sequences better, with **CrossNovo performing best overall**. We will include this line graph in our final version!
>
>
> **Thank you again for your effort and let us know if there's further questions!**

---

> > ### Comment · Reviewer_nsFd · 2025-08-02
> > **Further questions**
> >
> > Thank you for the author's response. The authors have mostly addressed my questions. Although there was no numerical display for Q4, it is still understandable.

---

> > > ### Author Response · Authors · 2025-08-02
> > > **Reply to Reviewer**
> > >
> > > **We thank the reviewer for their time and effort in reviewing our work again!** A line graph with clear length dependent performance will be included in our manuscript at revision as we promised and we thank the reviewer for the understanding!!
> > > best,
> > >  Authors

---

### Official Review · Reviewer_rxjr · 2025-06-29

**Clarity:** 3
**Significance:** 2
**Originality:** 2
**Rating:** 4
**Confidence:** 3

**Summary:**

The author proposed a novel framework combining both autoregressive (AT) and non-autoregressive (NAT) tranformers for de novo peptide sequencing. The two-stage training firstly pretrain both AT and NAT decoder jointly, and then finetune the AT decoder with NAT features to improve the overall performance of the AT model. The proposed CrossNovo model achieves strong performance compared with baselines.

**Questions:**

1. Related work should probably not include the model proposed in this work (line 92-97).
1. The problem formulation is clear but can be improved for readers who are not familiar with the domain knowledge. From my understanding, the length of the amino acid sequence $\mathbf{A}$ to be predicted is unknown, which is why AT is preferred over NAT method for this problem. It would be nice if the authors can mention that.
1. The authors mention augmenting the AT decoder with prefix and suffix mass information at each generation step (line 140-143). Can the authors elaborate?
1. Figure 3 and Figure 6: missing x and y axis labels.
1. Ablation study can be improved. First of all, the author should elaborate more about different ablation configurations. The shared encoder config is not explained clearly in the paper. More importantly, I am specifically curious about one possible configuration: if you train the NAT decoder first with spectrum encoder, and then train the AT decoder from scratch as in the second stage of training (cross attending to both spectrum encoder and NAT decoder features), how would the performance be. To me, the NAT decoder functions more as another “encoder” or “feature extractor” that provides bi-directional features to the AT decoder, hence I’m curious about the ablation of such configuration.
1. Can authors provide comparison of number of parameters of the proposed CrossNovo model and the baselines (such as CasaNovo V2 and ContraNovo)?

**Ethical Concerns:**

["NO or VERY MINOR ethics concerns only"]

**Final Justification:**

Interesting work that organically combines Autoregressive and Non-Autoregressive generation for specific domain application. Suggest to accept.

**Limitations:**

Yes

**Paper Formatting Concerns:**

See *Questions* section for some minor points.

**Quality:**

3

**Strengths And Weaknesses:**

**Strengh**
1. Clear explanation of the proposed model architecture.
1. Strong performance in benchmark compared to comprehensive list of baselines.

**Weakness**
1. Discussion of baselines is lacking for readers who are not familiar with the specific task.
1. Ablation study can be improved.
1. Formatting of manuscript and figures can be improved.
See *Questions* section for more details about weakness.

---

> ### Author Rebuttal · Authors · 2025-07-30
>
> # **Reviewer Comment:**
> > *“Discussion of baselines is lacking for readers who are not familiar with the specific task.”*
>
> **Author Response:**
> We thank the reviewer for pointing this out. Our original intention was to provide a detailed description of each baseline method in the main Experiments section. However, due to the strict page limit, we condensed the descriptions into a brief summary **(lines 246–255)** to prioritize core model designs.
>
> We agree that for readers unfamiliar with the de novo peptide sequencing domain, a more comprehensive explanation of the baselines would improve clarity and context. In the revised version of the paper, we will add a detailed comparison and methodological overview of each baseline in the `Appendix`. This will include their architectural differences, design motivations, and key performance characteristics to better situate our our contributions relative to prior work.
>
> # **Reviewer Comment:**
> > *“The problem formulation is clear but can be improved for readers who are not familiar with the domain knowledge. From my understanding, the length of the amino acid sequence to be predicted is unknown, which is why AT is preferred over NAT method for this problem. It would be nice if the authors can mention that.”*
>
> **Author Response:**
> We thank the reviewer for this helpful observation—your understanding is exactly correct.
>
> As we briefly noted in the paper, a key strength of the AT approach lies in its natural ability to generalize to sequences of arbitrary and unseen lengths. `This is particularly important for de novo peptide sequencing, where the length of the amino acid sequence is not known a priori`. In contrast, **NAT** models, such as PrimeNovo, must predefine a maximum generation length ( 40 tokens in their setting). As a result, they are unable to generate valid sequences longer than this limit, leading to failure cases in important long-peptide scenarios.
>
> Additionally,` AT models benefit from simpler and more stable optimization during training`. NAT models—especially those trained with CTC or other alignment-based objectives—often suffer from convergence issues and sensitivity to initialization and hyperparameters.
>
> We will revise the **Introduction** to explicitly state these advantages of AT models, both to better motivate our hybrid design and to improve clarity for readers who may be less familiar with the unique challenges of this domain.
>
> # **Reviewer Comment:**
> > *“The authors mention augmenting the AT decoder with prefix and suffix mass information at each generation step (lines 140–143). Can the authors elaborate?”*
>
> **Author Response:**
> Thank you for this question—this is an important design detail that we are happy to clarify.
>
> In de novo peptide sequencing, each input spectrum includes a known **precursor mass**, which represents the total molecular mass of the peptide being decoded. To help the model make more informed predictions and remain within the valid mass constraint, we augment the decoder at each generation step with two auxiliary signals:
>
> - **Prefix mass**: the cumulative mass of all previously predicted amino acids.
> - **Suffix mass**: the remaining mass, calculated as `(precursor mass – prefix mass)`.
>
> Providing this mass information helps the model reason about how much mass budget is left when selecting the next amino acid, reducing the chance of invalid or over-extended predictions.
>
> These values are continuous and domain-specific, so we encode them using **sinusoidal positional encoding** (adapted to float values) before injecting them as additional inputs at each decoding step. `While detailed implementation is available in our released cod, the motivation is to give the model explicit biochemical context for more accurate and mass-consistent decoding.`
>
> We will clarify this design choice more explicitly in the revised manuscript for better transparency and understanding!
>
> # **Reviewer Comment:**
> > *“Figure 3 and Figure 6: missing x and y axis labels.”*
>
> **Author Response:**
> We apologize for this oversight. Figures 3 and 6 are **Recall-Coverage curves**, where:
>
> - The **x-axis** represents the **percentage of prediction coverage**, determined by filtering predictions based on confidence scores.
> - The **y-axis** represents the **peptide-level recall** (also referred to as peptide precision in the main text).
>
> We greatly appreciate the reviewer’s careful reading and attention to detail. We will revise both figures in the final version to clearly include axis labels.
>
> # **Reviewer Comment:**
>
> > *Ablation study can be improved. First of all, the author should elaborate more about different ablation configurations. The shared encoder config is not explained clearly in the paper. More importantly, I am specifically curious about one possible configuration: if you train the NAT decoder first with spectrum encoder, and then train the AT decoder from scratch as in the second stage of training (cross attending to both spectrum encoder and NAT decoder features), how would the performance be. To me, the NAT decoder functions more as another “encoder” or “feature extractor” that provides bi-directional features to the AT decoder, hence I’m curious about the ablation of such configuration.*
>
> **Author Response:**
> We thank the reviewer for this insightful and thoughtful comment. We agree that the ablation study could benefit from more elaboration, and we appreciate the opportunity to clarify and expand on our configurations.
>
> In our current ablation (Table 4), the **“shared encoder”** configuration refers to removing the shared spectrum encoder and instead assigning independent encoders to the AT and NAT branches. This design assumes that AT and NAT decoders might focus on different aspects of the input spectrum and thus benefit from specialized encodings. In this setup, the only pathway for information exchange between NAT and AT is the **cross-decoder attention** mechanism. Notice that, `when both the shared encoder and cross-decoder connection are removed, the two decoders become completely disconnected`, which does not entail any row in Table 4 (i.e. either Cross-Decoder or Shared-Encoder needed to be included  .
>
> We also thank the reviewer for the **very interesting suggestion** regarding a staged training scheme where the NAT decoder is trained fully first as a feature extractor, and then the AT decoder is trained from scratch by cross-attending to both the fixed spectrum encoder and NAT features. We share the reviewer's view that the NAT decoder in our framework effectively acts as an auxiliary information encoding unit, providing enriched bi-directional representations of potential sequence. This configuration could lead to improved stability or representation quality, especially if the NAT branch is allowed to mature independently before the AT training begins.
>
> Due to the substantial computational demands of training de novo sequencing models (our training involves over 30 million spectra and requires more than **1,600 A100 GPU hours** for each configuration to be trained), we initially prioritized the most structurally distinct configurations for the ablation in Table 4. Some of these, like Row 2, reflect early-stage development result. But highly expensive computational cost prevents us to try all configs.
>
> That said, we fully agree with the reviewer’s intuition, and the proposed experiment aligns well with our training design. We can simulate this configuration by modifying the **importance annealing schedule** in our paper: instead of gradually decreasing the weight of the NAT loss (λ), we would fix the NAT loss at 100% for the initial phase (say first 50 epoch), then gradually introduce the AT loss later—effectively staging the training in the way the reviewer suggests.
>
> We have started preparations to run this variant and will include the results in the final version of the paper once it's finished. We again thank the reviewer for this valuable suggestion!
>
> # **Reviewer Comment:**
> > *Can authors provide comparison of number of parameters of the proposed CrossNovo model and the baselines (such as CasaNovo V2 and ContraNovo)?*
>
> **Author Response**:
> We thank the reviewer for raising this important point. Below we provide a comparison of model sizes between our proposed model and the main baselines:
>
> - **CrossNovo (AT only)**: 33 million parameters
> - **CROSSNOVO (AT + NAT total)**: 45 million parameters
>   - 9 Transformer layers per decoder, 400 hidden dimensions
>   - `NAT decoder serves as an auxiliary module for contextual feature extraction during training and cross-attention`
>
> For comparison:
> - **PrimeNovo (NAT)**: 33 million parameters
>   - Same number of layers and hidden dimensions as our NAT component
> - **Casanovo V2 (AT)**: ~38 million parameters
>   - 9 layers, 512 hidden dimensions
> - **ContraNovo (AT)**: ~43 million parameters
>   - 9 layers, 512 hidden dimensions, plus an additional contrastive module
>
>
>
> We hope this comparison provides the information the reviewer needs, we will make this further clear in the revision!
>
>
> **We sincerely thank the reviewer for the thoughtful and detailed feedback, and we are happy to address any further questions if you have!**

---

### Official Review · Reviewer_2Zcz · 2025-07-02

**Clarity:** 3
**Significance:** 3
**Originality:** 3
**Rating:** 4
**Confidence:** 3

**Summary:**

This study addresses the de novo peptide sequencing problem in mass spectrometry-based proteomics, a critical challenge in proteomic research. Prior approaches have relied primarily on autoregressive (AR) or non-autoregressive (NAR) models, which face limitations in capturing bidirectional sequence dependencies (AR) or ensuring generative coherence and scalability (NAR). To overcome these issues, we introduce a hybrid framework integrating AR and NAR paradigms through: (1) A shared spectral encoder coupled with dual decoders (AR and NAR) featuring cross-decoder attention mechanisms to leverage complementary strengths. (2) Importance annealing to dynamically balance contributions from the two decoding strategies and cross-decoder gradient blocking to prevent interference during training.
This study provides validation across a 9-species benchmark dataset and antibody-specific peptide datasets, demonstrating significant performance improvements over state-of-the-art methods.

**Questions:**

1. Previous studies have explored hybrid autoregressive (AR) and non-autoregressive (NAR) decoders for various traditional NLP tasks, such as speech recognition. However, the unique advantages or innovations of this hybrid approach in the specific context of peptide sequencing remain unclear. What distinct features does the proposed method offer to address the challenges inherent to peptide sequencing that set it apart from existing hybrid frameworks?
2. Extending the analysis to post-translationally modified (PTM) peptides—similar to the work in PrimeNovo (https://d-nb.info/1360284540/34)—would be a valuable addition. Investigating how the method performs on PTM peptides could reveal its applicability to more complex proteomic scenarios.
3. Attention map and feature vector similarity study would benefit the study.

**Ethical Concerns:**

["NO or VERY MINOR ethics concerns only"]

**Final Justification:**

The reviewers did a nice job and addressed all my concerns.

**Limitations:**

In summary, the paper’s key strength lies in its presentation of an effective approach that integrates autoregressive (AR) and non-autoregressive (NAR) architectures within a single model, leading to significant performance gains. A notable weakness, however, is the lack of clarity regarding the uniqueness or novelty of this hybrid AR/NAR framework when applied specifically to the peptide sequencing task.

**Paper Formatting Concerns:**

I have no formatting concerns

**Quality:**

3

**Strengths And Weaknesses:**

Strengths:
1. The method undergoes rigorous evaluation and demonstrates significant performance improvements in de novo peptide sequencing tasks.
2. The study is overall well-written and clearly presented.
3. This study can benefit the audience in the peptide sequencing domain.

Weakness:
1. Previously, hybrid autoregressive and non-autoregressive decoder was studied by many other papers for various traditional NLP tasks, including speech recognition. It is not clear what is the uniqueness of the method for peptide sequencing task?
2. It would be interesting to look into the post-translational modified peptide task as PrimeNovo did (https://d-nb.info/1360284540/34)

---

> ### Author Rebuttal · Authors · 2025-07-30
>
> # **Reviewer Comment:**
>
> Weakness:
> > *“Previously, hybrid autoregressive and non-autoregressive decoder was studied by many other papers for various traditional NLP tasks, including speech recognition. It is not clear what is the uniqueness of the method for peptide sequencing task?”*
>
> Questions:
> > What distinct features does the proposed method offer to address the challenges inherent to peptide sequencing that set it apart from existing hybrid frameworks?
>
>
> **Author Response:**
> We appreciate the reviewer’s insight and the opportunity to clarify the novelty and motivation of our hybrid design in the context of peptide sequencing. To provide a comprehensive response, we will take a step back to dive into the details of our initial design motivations and rationale below.
>
> In natural language tasks, information typically flows uni-directionally — for example, in translation, summarization, or dialogue systems, where earlier context is usually sufficient for predicting the next token. This aligns with how humans generate and interpret language sequentially (from left to right), and as a result, **autoregressive (AT)** models like GPT and LLaMA dominate performance benchmarks across NLP tasks. **Non-autoregressive (NAT)** models in NLP, while explored, are predominantly motivated by inference speed, not accuracy, and they generally underperform AT models unless substantial constraints or post-processing are applied (e.g. [Liu et al. 2022] ).
>
> Even when hybrid approaches are proposed (e.g., Mask-Predict or iterative refinement or joint decoder learning in NLP), these often use NAT components to improve efficiency or as auxiliary objectives (e.g., Hao et al., 2020) often with no actual bi-directional information directly being accessed during prediction, as they don't provide additional performance and information gain due to already powerful AT modelling power in natural language as stated above.
>
> Biological sequences, however, differ fundamentally from language. Each token (amino acid) in a peptide does not have a fixed or dominant direction of information flow. Instead, `its identity and functional implications often depend on both the residues front and after`. This is evident from the prevalence of bi-directionally trained protein language models (e.g., ESM-2, ESM-3), which have achieved state-of-the-art performance in many protein-related tasks by leveraging full-sequence context (even generational tasks, where ESM generates accurate protein sequences in given regions using NAT architecture).
>
> In the**de novo peptide sequencing task**, the NAT model, such as PrimeNovo, also outperforms AT baselines in accuracy as shown by previous study, partially because it enables `full bidirectional context`, **which is naturally aligned with the underlying biological dependencies** in spectral data. However, we found that NAT models face challenges in length generalization, training stability, and scaling to longer sequences due to architectural constraints and reliance on fixed-length decoding (PrimeNovo predefines generation length up to 40, while AT allows much longer in our experiments).
>
> Our contribution is to bridge these complementary strengths __(AT's scalability, training stability and higher performance in certain species like human, NAT's bi-directional features)__ through a novel and *task-specific* hybrid architecture: by introducing a dedicated NAT decoder to learn bidirectional latent features **direction**, and a separate AT decoder that performs sequence generation by attending to these features via a **cross-decoder attention mechanism**. This attention mechanism **directly** integrates the bidirectional NAT representations into the AT decoding steps, enhancing prediction accuracy while retaining the generative flexibility, optimization benefits, and length generalization of AT models.
>
> To the best of our knowledge, this is the first work to introduce such a direct, modular, and fine-grained information flow from a NAT decoder into an AT decoder for sequence generation. **Unlike prior hybrid approaches in NLP or speech, where NAT modules are often auxiliary or limited to inference-time ensembling and speed improving** (as autoregressive already performs better than non-autoregressive), our model `transfers the bidirectional context directly into AT decoding as a core predictive mechanism` for `performance gain`. We believe this bi-directional information transfer paradigm is particularly beneficial for biological sequences, where no strict left-to-right generational assumption holds.
>
> As a result , while hybrid AT/NAT ideas exist in NLP,  the motivation and implementation in our work are biologically motivated and architecturally different and novel, tailored to the fundamental differences in peptide sequences compared to natural language. We hope this resolves your concerns with regards to our hybrid design and we will further make this clear in our manuscript during revision!
>
>
> # **Reviewer Comment:**
> > *“Extending the analysis to post-translationally modified (PTM) peptideswould be a valuable addition. ”*
>
> **Author Response**
> We sincerely thank the reviewer for this valuable suggestion. In response, we have extended our analysis to include post-translationally modified (PTM) peptides following the setup described in the PrimeNovo paper.
>
> Specifically, we augmented our amino acid vocabulary to include additional PTM tokens and fine-tuned our pretrained **CrossNovo** model on the same **21PTM** dataset used in PrimeNovo. Due to the limited rebuttal time, we  finetuning our model on each one of the first five PTMs (in alphabetical order): **Acetylation**, **Biotinylation**, **Crotonylation**, **Butyrylation**, and **Dimethylation**, following settings in PrimeNovo.
>
> The following table presents side-by-side comparisons between **PrimeNovo** and **CrossNovo** on these PTMs across three metrics: **Classification Accuracy**, **Amino Acid Recall**, and **Peptide Recall**, where PrimeNovo numbers are quoted from their paper and CrossNovo are fine-tuned on each one of the PTM data split using the same setting as PrimeNovo paper.
>
> | **Metric**                  | **Model**   | **Acetylation** | **Biotinylation** | **Crotonylation** | **Butyrylation** | **Dimethylation** |
> |----------------------------|-------------|------------------|-------------------|--------------------|------------------|--------------------|
> | **Classification Accuracy** | PrimeNovo   | 0.98             | **0.99**          | 0.98               | 0.97             | 0.95               |
> |                            | CrossNovo   | **0.98**         | 0.98              | **0.98**           | **0.98**         | **0.97**           |
> | **AA Recall**              | PrimeNovo   | 0.95             | **0.83**          | **0.90**           | 0.82             | 0.84               |
> |                            | CrossNovo   | **0.96**         | **0.83**          | 0.86               | **0.85**         | **0.85**           |
> | **Peptide Recall**         | PrimeNovo   | 0.89             | 0.70              | **0.79**           | 0.65             | 0.66               |
> |                            | CrossNovo   | **0.90**         | **0.72**          | 0.78               | **0.69**         | **0.68**           |
>
> As shown above, CrossNovo achieves competitive or improved performance in nearly all categories, particularly **Peptide Recall**, which are critical for accurate peptide sequence prediction.
>
> Due to the short rebuttal window we were only able to evaluate 5 of the 21 PTMs (chosen based on the aliphatic order). Still, the consistent performance improvements suggest that our approach can serve as a strong foundation for handling PTM peptides in more complex proteomic scenarios. We plan to extend the evaluation to all 21 PTMs after rebuttal to further strengthen our findings. Thank you again!
>
> # **Reviewer Comment:**
> > *“Attention map and feature vector similarity study would benefit the study.”*
>
> **Author Response:**
> We thank the reviewer for this insightful suggestion.  We performed an analysis of attention patterns in CrossNovo's AT decoder and compared them to those observed in Contra. and Casa. AT decoder.
>
> `While the rebuttal policy prevents us from including figures or external links, we summarize our findings here`:
>
> In Casa, we found that a significant proportion of attention weight is narrowly concentrated on the first token of the input, similar to what PrimeNovo noticed.  A better behaved attention pattern was noted in ContraNovo with more diversity across positions in input.
>
> In comparison, our model demonstrates a much more diverse and context-sensitive attention pattern. The attention is distributed across wider range of peak inputs and more often allocated to spectral positions corresponding to peaks that are b- or y-ion fragments.
>
> We also analyzed the **cross-decoder attention map**, which enables the AT decoder to incorporate contextual knowledge from the NAT decoder. Interestingly, we observed a consistent pattern: when the AT decoder generates a token at position *t*, its attention is predominantly aligned with the rough predictive region of t-th token position in the NAT feature map. This alignment suggests that the AT decoder has learned to focus on the NAT’s bidirectional contextual embedding at the corresponding position, leveraging it as a complementary signal during generation. We believe these observations offer some more insights in the effectiveness of designs and we will further discuss in the revision, thank you again for this!
>
> **We thank the reviewer again for the  constructive feedback. Please feel free to let us know if there are any further questions or concerns!!**
>
> # Reference
>
> Liu et al., (2022, May). Learning Non-Autoregressive Models from Search for Unsupervised Sentence Summarization.
>
>
> Hao et al. (2020) Multi-task learning with shared encoder for non-autoregressive machine translation.

---

### Note · Authors · 2025-08-12

We sincerely thank all reviewers for their insightful and constructive feedback. We are encouraged that reviewers found our work **"well-written and clearly presented" (2Zcz)**, with a **"sophisticated and mathematically rigorous framework" (7Vk1)**, **"strong performance" (rxjr)**, and **"potential for practical biological research" (nsFd)**.

We have carefully considered all comments and have conducted additional experiments during the rebuttal period. Below, we summarize our rebuttal and planned revisions.

---

The reviewers' most pointed feedback centered on  clarifications on methodological details and hybrid designs.

**1. Design and Motivation of the Hybrid Architecture:**

 We clarified that unlike in NLP where hybrid models often target speed, our design is biologically motivated to improve **accuracy**. Peptides have inherent bidirectional dependencies that pure AT models miss. Our core innovation allows the AT model to leverage global context during its sequential generation.

**2. Additional Experimental Validation:** Reviewers requested several new experiments:

**Our Reply:**
* **PTM Peptides:** We fine-tuned and evaluated our model on 5 PTM datasets from PrimeNovo, demonstrating **competitive or superior performance**.
* **Qualitative Analysis:** As requested (2Zcz, nsFd), we analyzed attention maps, finding that our model learns more diverse patterns and that the AT decoder effectively attends to corresponding regions in the NAT feature map, validating our design.
* **Computational Cost:** We provided concrete benchmarks, showing our model has a comparable number of parameters to baselines (45M vs. 38M-43M) with a moderate increase in inference latency due to cross-attention mechanism.
* **Ablations & Sensitivity:** We clarified that the extreme computational cost (~1600 GPU-hours per run) makes exhaustive hyperparameter sweeps infeasible, a common constraint in this field. However, we have begun running the insightful staged-training ablation and will include the results.

--

**Commitment for Final Version**

We are committed to incorporating all feedback to significantly improve the manuscript.  We will revise the introduction to better frame the problem  and conduct a thorough proofread to fix all formatting and referencing issues.



We believe these revisions will address all raised concerns and substantially strengthen the paper. We are confident in our contributions and thank the reviewers and AC again for their valuable guidance.

---

### Decision · Program_Chairs · 2025-09-17

**Decision:**

Accept (poster)

**Comment:**

The paper presents CrossNovo, a hybrid framework for de novo peptide sequencing that integrates the strengths of autoregressive (AT) and non-autoregressive (NAT) models. The approach employs a shared spectrum encoder with dual decoders, where the NAT decoder captures bidirectional context and the AT decoder incorporates these features through cross-decoder attention, supported by importance annealing and gradient blocking. Reviewers consistently highlighted the strong empirical performance, the clarity of the architectural design, and the significance of the contributions. Concerns raised regarding missing baselines, prior work on hybrid models in NLP and speech, and computational cost were largely addressed during the rebuttal. Overall, this is a solid and impactful contribution, and I recommend acceptance.